# AdaCoT: Pareto-Optimal Adaptive Chain-of-Thought Triggering via Reinforcement Learning

## Abstract

Large Language Models (LLMs) have demonstrated remarkable capabilities but often face challenges with tasks requiring sophisticated reasoning. While Chain-of-Thought (CoT) prompting significantly enhances reasoning, it indiscriminately generates lengthy reasoning steps for all queries, leading to substantial computational costs and inefficiency, especially for simpler inputs. To address this critical issue, we introduce AdaCoT (Adaptive Chain-of-Thought), a novel framework enabling LLMs to adaptively decide when to invoke CoT. AdaCoT framed adaptive reasoning as a Pareto optimization problem that seeks to balance model performance with the costs associated with CoT invocation (both frequency and computational overhead). We propose a reinforcement learning (RL) based method, specifically utilizing Proximal Policy Optimization (PPO), to dynamically control the CoT triggering decision boundary by adjusting penalty coefficients, thereby allowing the model to determine CoT necessity based on implicit query complexity. A key technical contribution is Selective Loss Masking (SLM), designed to counteract decision boundary collapse during multi-stage RL training, ensuring robust and stable adaptive triggering. Experimental results demonstrate that AdaCoT successfully navigates the Pareto frontier, achieving substantial reductions in CoT usage for queries not requiring elaborate reasoning. For instance, on our production traffic testset, AdaCoT reduced CoT triggering rates to as low as 3.18% and decreased average response tokens by 69.06%, while maintaining high performance on complex tasks. This substantial token decrease directly translates to a significant reduction in inference computational load. AdaCoT pioneers adaptive CoT triggering, offering a practical and principled solution for developing more efficient, responsive, and cost-effective LLMs, particularly crucial for interactive and resource-sensitive applications.

## 1 Introduction

Large Language Models (LLMs) have garnered substantial attention due to their remarkable ability to encode extensive world knowledge from vast corpora [1], enabling impressive performance across diverse tasks such as question answering, creative writing, and summarization. Despite these successes, LLMs often demonstrate limitations in tasks requiring sophisticated reasoning, including solving complex mathematical problems and intricate coding puzzles. To mitigate this, recent methodologies have employed Chain-of-Thought (CoT) prompting [2], which encourages models to explicitly generate step-by-step reasoning prior to producing final answers. This approach significantly enhances the reasoning capability of models, even achieving human-expert levels in certain domains [3, 4, 5, 6].

Submitted to 39th Conference on Neural Information Processing Systems (NeurIPS 2025). Do not distribute.

However, employing CoT prompting also poses critical challenges during inference [7]. Specifically, it substantially increases the number of tokens generated, even for simple queries that do not benefit from elaborate reasoning, such as straightforward arithmetic questions. This indiscriminate token expense consequently raises deployment costs and reduces inference efficiency. Ideally, a model should adaptively determine when detailed reasoning is necessary. For instance, simple queries like "What is 1+1?" should be answered immediately without additional reasoning steps, whereas more complex queries require deeper and step-by-step reasoning. An adaptive strategy would thus optimize token usage, balancing cost-efficiency with response quality.

Recently, a few efforts has made attempts towards this direction. These approaches can be broadly categorized into three main directions. (1) Incorporate length penalties or brevity rewards during the reinforcement learning (RL) stage to encourage shorter, more concise reasoning paths [8, 9, 10, 11, 12, 13]. (2) Restructure CoT outputs through post-processing such as iterative summarization or pruning [14, 15, 16, 17, 18, 19, 20, 21]. (3) Employ explicit user instructions or hand-crafted selection mechanisms to control the use of CoT [22, 23, 24, 25, 26, 27]. Despite their contributions, they mainly focus on monotonic reasoning reduction, failing to account for the nuanced variability in query complexity, i.e., treating simple and difficult prompts adaptively. Moreover, they lack a principled optimization framework to guide balancing response quality against deployment cost.

To address these limitations, we introduce AdaCoT (Adaptive Chain-of-Thought), a novel approach grounded in formal mathematical analysis. Our key insight is framing adaptive reasoning as a multi-objective optimization problem with two competing goals: maximizing response accuracy and minimizing deployment costs. Specifically, we formalize this balance through Pareto optimization, seeking optimal trade-offs between reasoning complexity and inference efficiency. Such a mathematical framework provides clear theoretical grounding for dynamically adapting CoT triggering based on query complexity.

Leveraging this formalization, we propose an RL-based strategy explicitly designed around the Pareto optimization framework, enabling effective control of the model's decision boundary for initiating CoT prompting. During training, the RL agent dynamically assesses the complexity of incoming user queries to determine the necessity and extent of reasoning steps. By carefully designing the reward function to incorporate penalty coefficients, we encourage the RL agent to seek solutions along the Pareto frontier, explicitly optimizing trade-offs between response accuracy and token expenditure. This structured exploration enables the model to effectively discern when detailed reasoning is beneficial, thereby systematically enhancing inference efficiency and significantly reducing deployment costs.

The proposed AdaCoT framework delivers substantial benefits in LLM operational efficiency. By empowering models to selectively engage CoT, AdaCoT can reduce triggering rates to as low as 3.18% and cut average response tokens by 69.1% in production settings. This significant reduction in computational load is achieved while maintaining strong performance on 15 widely-adopted benchmarks. For example, AdaCoT can achieve a 62.8% average score using only a 53.3% CoT rate, closely rivaling the 65.0% score of a model that always employs CoT. These improvements directly translate to more cost-effective and responsive LLM systems.

## 2 The AdaCoT Framework

Enabling a large language model (LLM) to dynamically decide whether to invoke Chain-of-Thought (CoT) reasoning based on the complexity of user queries is a critical task, which allows LLMs to allocate computational resources more rationally, i.e., spending tokens on complex reasoning tasks while avoiding unnecessary overhead for simple queries.

To achieve this, we introduce AdaCoT, a unified framework for adaptive reasoning. The central insight is that the decision to employ CoT prompting can be naturally cast as an optimization problem involving two competing goals: (1) maximizing response accuracy and (2) minimizing deployment costs. To capture this trade-off, we frame the task as a Pareto optimization problem, providing a principled foundation for balancing reasoning quality against computational efficiency. Based on this formulation, we propose an RL-based control strategy that governs the invocation of CoT reasoning. Specifically, we train a policy model learning to assess the complexity of each query and decide whether CoT reasoning should be used. During RL training, the policy model is optimized using a reward signal carefully designed to reflect the Pareto trade-off, incorporating both task performance

and token efficiency. Through this RL-based mechanism, AdaCoT learns to allocate reasoning effort adaptively, yielding high-quality responses with minimal computational cost.

## 2.1 Adaptive Reasoning as a Pareto Optimization Problem

We formulate the adaptive reasoning challenge as a Pareto optimization problem, aiming to simultaneously maximize model performance and minimize CoT usage. Let $\mathcal{D} = \{(x_i, y_i)\}_{i=1}^{N}$ be a dataset of query-response pairs, where $x_i$ is the input query and $y_i$ is the ground truth response. Let $f_\theta$ be an LLM parameterized by $\theta$, and let $r_\theta(x_i)$ be the response generated by the model for input $x_i$.

To CoT usage is measured by the CoT triggering rate $T(\theta)$, defined as the proportion of responses that include reasoning:

$$T(\theta) = \frac{1}{N} \sum_{i=1}^{N} \mathbf{1}[\text{HasReasoning}(r_\theta(x_i))] \tag{1}$$

where $\mathbf{1}[\cdot]$ is the indicator function and $\text{HasReasoning}(\cdot)$ determines if a response contains explicit CoT steps (e.g., non-empty content within `<think>...</think>` tags).

On the other hand, model performance $P(\theta)$ is defined as the average score on a set of evaluation metrics:

$$P(\theta) = \frac{1}{M} \sum_{j=1}^{M} \text{Score}_j(\theta) \tag{2}$$

where $M$ is the number of evaluation instances/metrics and $\text{Score}_j(\theta)$ is the model's score on the $j$-th evaluation.

Putting CoT usage $T(\theta)$ and model performance $P(\theta)$ together, the objective is to find model parameters $\theta^*$ that achieve an optimal trade-off:

$$\theta^* = \arg\max_\theta \{\lambda_P \cdot P(\theta) - \lambda_T \cdot T(\theta)\} \tag{3}$$

or more generally, to find solutions on the Pareto frontier of $(P(\theta), 1 - T(\theta))$. The hyperparameters $\lambda_P$ and $\lambda_T$ (or their implicit ratio) control the relative importance of performance versus CoT reduction. AdaCoT focuses on providing mechanisms to explore this frontier.

## 2.2 Training Pipeline for AdaCoT

The AdaCoT training pipeline integrates supervised fine-tuning (SFT) as an initialization phase, followed by multi-stage reinforcement learning (RL) to refine the adaptive CoT triggering behavior.

### 2.2.1 Data Preparation and Supervised Fine-Tuning (SFT) as Warm-up

To provide the model with an initial understanding of when CoT might be beneficial, we perform a data preparation stage. This is achieved by leveraging an auxiliary model, guided by a set of predefined principles (e.g., query complexity, expected reasoning depth, domain; see Appendix B). In our implementation, we use an internal 15B-parameter model to generate these annotations; however, the framework is model-agnostic and can be instantiated using any sufficiently capable LLM with basic instruction-following abilities. Queries are labeled as either likely benefiting from CoT or likely suitable for a direct answer. This principled, automated labeling process is more consistent and scalable than manual annotation.

The SFT training data is then structured as follows: For queries labeled as benefiting from CoT, responses retain the full reasoning process: `<think>reasoning_steps</think>answer`. For queries labeled as not requiring CoT, responses omit explicit reasoning but maintain structural consistency: `<think></think>answer`. This SFT stage serves as a "warm-up", equipping the model with a foundational capability to distinguish between these two response styles. The consistent use of `<think></think>` tags is crucial for maintaining response format integrity.

### 2.2.2 Reinforcement Learning for Adaptive CoT Control

The RL stage is pivotal for fine-tuning AdaCoT's adaptive reasoning capabilities. We design a reward function $R(x, r)$ for an input query $x$ and generated response $r$:

$$R(x, r) = R_{\text{base}}(x, r) - \alpha_1 \cdot P_{\text{miss}}(x, r) - \alpha_2 \cdot P_{\text{over}}(x, r) - \gamma \cdot P_{\text{fmt}}(r) \tag{4}$$

where $R_{\text{base}}(x, r)$ is the base reward reflecting response quality, $P_{\text{miss}}(x, r)$ is a binary penalty for reasoning omission, $P_{\text{over}}(x, r)$ is a binary penalty for reasoning overuse, $P_{\text{fmt}}(r)$ is a binary penalty for format errors, and $\alpha_1, \alpha_2, \gamma$ are non-negative penalty coefficients. By adjusting $\alpha_1$ and $\alpha_2$, we steer AdaCoT towards different CoT triggering decision boundaries, allowing exploration of the Pareto frontier.

### 2.2.3 Addressing Decision Boundary Collapse with Selective Loss Masking

A significant challenge in multi-stage RL, particularly when fine-tuning on specialized datasets with skewed CoT distributions (e.g., mathematical datasets where CoT is almost always beneficial), is the risk of the adaptive CoT triggering capability becoming unstable or collapsing. The model might revert to a homogeneous behavior, either always or never triggering CoT, thereby losing the nuanced decision-making learned in earlier, more balanced training stages. We term this phenomenon decision boundary collapse. This is particularly problematic if the final RL stage has significant bias, as it can lead to the model almost completely losing its adaptive triggering capability.

To address decision boundary collapse, AdaCoT incorporates Selective Loss Masking (**SLM**). SLM aims to preserve the CoT triggering ratio and distribution established during SFT or prior RL stages. It achieves this by selectively masking the loss contribution from the pivotal "decision token" during RL phases prone to distribution bias. This decision token is defined as the token immediately succeeding the `<think>` tag.

The modified policy gradient loss under SLM, $\mathcal{L}_{\text{SLM}}$, is computed by excluding the loss component associated with this decision token:

$$\mathcal{L}_{\text{SLM}} = \sum_{k \neq k_{\text{decision}}} \ell_k \tag{5}$$

where $\ell_k$ is the original loss component for the $k$-th token, and $k_{\text{decision}}$ is the index of the decision token.

## 3 Experiments

We conducted extensive experiments to evaluate the AdaCoT framework, focusing on its ability to navigate the performance-cost trade-off, the effectiveness of its adaptive triggering mechanism, and its impact on inference efficiency. This section details our experimental setup, presents the main results, and analyzes the findings.

### 3.1 Experimental Setup

For our base model, we utilized our internal 15B/150B parameter Mixture-of-Experts (MoE) [28, 29] model. The AdaCoT post-training process comprised an initial SFT stage, followed by a two-stage RL procedure: first, a Mathematics-Focused RL stage (RL-Math) concentrated on complex, rule-verifiable problems, and second, a General Domain RL stage (RL-General) which incorporated broader data and a trained reward model. We compared our **AdaCoT RL Models** (Exp1-Exp4) against several baselines: a **Full CoT SFT Baseline** (SFT model always generating CoT), a **Full CoT RL Baseline** (RL model derived from the Full CoT SFT, always generating CoT), a **No CoT SFT Baseline** (SFT model never generating CoT), a **No CoT RL Baseline** (RL model derived from the No CoT SFT, never generating CoT), and our **AdaCoT SFT Model** (our model after only the SFT stage, also referred to as Adaptive SFT Model).

The SFT and RL training datasets were constructed to cover a diverse range of domains, including mathematics, reasoning, professional disciplines (e.g., law, medicine), dialogue, creative writing, and general knowledge question answering. Both SFT and RL data were labeled for CoT necessity using the principle-guided assessment detailed in Appendix B. In the SFT dataset, approximately 67% of the samples were labeled as requiring CoT, while in the RL dataset, this proportion was around 40%. During SFT, queries identified as not requiring CoT were formatted with empty `<think></think>` tags. In the RL-Math stage, which is particularly prone to decision boundary collapse, we employed Selective Loss Masking (SLM), as described in Section 2.2.3. For the RL-General stage, we applied penalties according to Equation 4, systematically varying the $\alpha_1$ and $\alpha_2$ coefficients to explore different points on the Pareto frontier. Proximal Policy Optimization (PPO) [30] was used for all RL policy updates.

For evaluation, we used 15 diverse open-source benchmark datasets to assess overall performance, measured by the average score. To balance internal iteration efficiency with evaluation accuracy, some of these datasets underwent up-sampling or down-sampling, or the number of inference runs per test sample was adjusted (with the final score being an average over multiple inferences). These datasets include LiveBench [31], MMLU Pro [32], SuperGPQA [33], GPQA [34], Chinese SimpleQA [35], SimpleQA [36], AIME24 & AIME25, MATH [37], OlympiadBench [38], SweBench Agentless [39], LiveCodeBench [40], KOR-Bench [41], ProcBench [42], and SysBench [43]. The detailed per-dataset scores, which form the basis for our average score calculations, are presented in Appendix A (Table 4). To specifically assess CoT triggering decisions on typical user queries, we curated a high-quality balanced test set of 1000 prompts. These prompts were labeled for CoT necessity using the same principle-guided assessment as our SFT/RL training data and subsequently underwent manual verification to ensure label accuracy. On this set, we report CoT Triggering Accuracy, F1-score, Precision, and Recall, where the positive class indicates that CoT is required. Other metrics included the CoT triggering rate on the benchmark datasets and the average response token num on production setting.

## 3.2 Results and Analysis

Our results demonstrate AdaCoT's ability to effectively control CoT invocation, leading to improved efficiency while maintaining strong performance.

### 3.2.1 Pareto Frontier Analysis

We trained four variants of our AdaCoT RL model (Exp1-Exp4) by varying the penalty coefficients $\alpha_1$ (for missing CoT) and $\alpha_2$ (for overusing CoT). The specific coefficients were: Exp1 ($\alpha_1 = 0.1, \alpha_2 = 0.3$), Exp2 ($\alpha_1 = 0.2, \alpha_2 = 0.3$), Exp3 ($\alpha_1 = 0.3, \alpha_2 = 0.3$), and Exp4 ($\alpha_1 = 0.3, \alpha_2 = 0.1$). The format error penalty $\gamma$ was consistently set to $1.0$. Figure 1 illustrates the average score plotted against the CoT triggering rate for these models and the baselines, based on the average performance across our 15 benchmark datasets (detailed in Appendix Table 4). The No CoT SFT baseline achieved an average score of 43.6% with 0% CoT usage, while the No CoT RL baseline improved this to 47.7% at 0% CoT. The AdaCoT SFT Model (Adaptive SFT) registered a 57.1% average score at a 61.3% CoT rate.

The AdaCoT RL models trace a compelling Pareto frontier. AdaCoT RL Exp1 (43.1% CoT, 59.7% score) and AdaCoT RL Exp2 (53.3% CoT, 62.8% score) demonstrate significant performance gains over the AdaCoT SFT model while operating at lower or comparable CoT rates. Notably, AdaCoT RL Exp2 achieves a 62.8% average score, approaching the Full CoT RL baseline (65.0% score, 100% CoT) with nearly half the CoT usage. As we increase the CoT triggering rate, AdaCoT RL Exp3 (65.4% CoT, 64.3% score) and AdaCoT RL Exp4 (67.7% CoT, 64.4% score) further push performance, closely rivaling the Full CoT RL baseline's score but with approximately 32-35% less CoT invocation. Moreover, these results indicate that despite the fixed CoT triggering labels within the SFT/RL data, adjusting the combination of penalty coefficients during the RL phase enables the final RL model to learn triggering strategies that transcend these initial labeling constraints.

This highlights AdaCoT's effectiveness in navigating the trade-off between performance and CoT cost. However, it is also observable that while the AdaCoT RL models achieve substantial efficiency gains and define a superior Pareto curve compared to simpler baselines, they encounter a slight performance bottleneck when their triggering rates are pushed higher. Specifically, even the highest-performing adaptive models (Exp3 and Exp4, with scores of 64.3% and 64.4% respectively) do not surpass the average score of the Full CoT RL baseline (65.0%). This suggests that while AdaCoT excels at reducing CoT for a vast majority of queries without compromising much on average performance, and indeed offers a better score-to-cost ratio, the absolute peak average performance achieved by a model specialized to always use CoT (Full CoT RL) remains marginally higher. This indicates that the adaptive mechanism, while highly effective, might not perfectly replicate or exceed the performance of an always-on CoT strategy in every single scenario contributing to the average, thus not fully crossing this specific optimal boundary for maximum average score. This could be due to the inherent complexities of learning a universally optimal triggering heuristic or the RL optimization finding a balance that prioritizes the significant cost savings available across the query distribution.

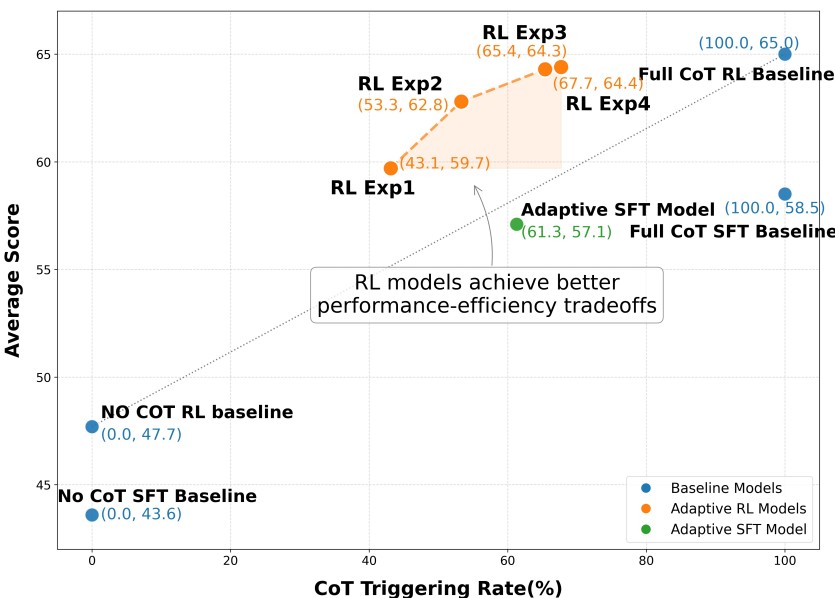

Figure 1: Average Score vs. CoT Triggering Rate across 15 widely-adopted benchmarks. Blue points represent baseline models. The green point is the AdaCoT SFT model. Orange points represent AdaCoT RL models trained with varying penalty coefficients, forming an improved Pareto frontier (indicated by the orange dashed line and shaded region) over the baselines. The dotted line connects the No CoT RL baselines to the Full CoT RL baseline, illustrating a simpler trade-off curve.

### 3.2.2 Adaptive CoT Triggering Performance and Ablation Studies on Daily-Use Queries

We evaluated the CoT triggering capabilities of AdaCoT at various training stages using our curated 1000-prompt daily-use test set. Table 1 presents these results, which include an ablation study for SLM and an assessment of the meta-reasoning strategy (discussed further in Section 4.2) at the SFT stage.

Table 1: CoT triggering performance on the 1000 daily-use prompt test set across different AdaCoT stages and configurations (positive class: requires CoT). RL-Math is the Mathematics-Focused RL stage; RL-General refers to the final models (Exp1-4).

| Model Stage / Variant | Accuracy | F1-Score | Recall | Precision |
|---|---|---|---|---|
| AdaCoT SFT Model | 0.795 | 0.750 | 0.616 | 0.959 |
| AdaCoT RL Model (Exp1 - RL-General) | 0.657 | 0.484 | 0.322 | 0.975 |
| AdaCoT RL Model (Exp2 - RL-General) | 0.816 | 0.814 | 0.804 | 0.823 |
| AdaCoT RL Model (Exp3 - RL-General) | 0.809 | 0.789 | 0.716 | 0.879 |
| AdaCoT RL Model (Exp4 - RL-General) | 0.678 | 0.535 | 0.370 | 0.963 |
| RL-Math (without SLM) | 0.506 | 0.669 | 1.000 | 0.503 |
| RL-Math (with SLM) | 0.813 | 0.781 | 0.670 | 0.938 |
| AdaCoT SFT Model (with Meta-Reasoning) | 0.858 | 0.840 | 0.762 | 0.935 |

The AdaCoT SFT model itself provides a strong baseline for adaptive triggering. The results clearly show that the RL-Math stage without SLM suffers from severe decision boundary collapse: the model defaults to triggering CoT (Recall=1.0) but with poor precision (0.503) and consequently low overall accuracy (0.506). Applying SLM during the RL-Math stage effectively preserves the adaptive capability learned during SFT, maintaining high precision (0.938) and achieving significantly better accuracy (0.813). The final AdaCoT RL models (Exp1-4, emerging from the RL-General stage) demonstrate how adjusting the RL penalty coefficients ($\alpha_1, \alpha_2$) allows for fine-tuning of the decision

boundary. AdaCoT RL Model Exp2, for example, achieves a well-balanced F1-score of 0.814. The incorporation of a meta-reasoning strategy at the SFT stage also shows a notable improvement in triggering performance, a point elaborated in Section 4.2.

### 3.2.3 Response Length Reduction and Efficiency Gains

The adaptive reasoning enabled by AdaCoT translates into significant reductions in computational costs. Table 2 shows the average response length and CoT triggering rates for AdaCoT RL Model Exp2 (selected for its balanced performance on the daily-use set and strong average benchmark performance) when applied to our production traffic testset, which reflects natural, unfiltered user query distributions.

Table 2: Average response token num (with reduction noted) and CoT triggering rate on production traffic testset for AdaCoT RL Model Exp2 vs. Full CoT RL Baseline.

| Platform | Model / Mode | Avg. Response Tokens | CoT Triggering Rate |
|---|---|---|---|
| Mobile | Full CoT RL Baseline | 377.18 | 100.00% |
| | AdaCoT RL Model Exp2 (Adaptive) | 116.70 (↓69.1%) | 3.18% |
| PC | Full CoT RL Baseline | 1376.31 | 100.00% |
| | AdaCoT RL Model Exp2 (Adaptive) | 405.25 (↓70.6%) | 12.50% |

As evidenced, AdaCoT RL Model Exp2 achieves very low CoT triggering rates in a production setting (3.18% on mobile devices, 12.50% on PCs). This dramatic reduction from the 100% CoT usage of a non-adaptive model translates directly into substantial computational savings.

## 4 Discussion and Future Work

### 4.1 Design Considerations and Limitations

AdaCoT offers a pragmatic approach to adaptive reasoning by combining principle-guided initial data labeling with RL-based optimization of the CoT decision boundary. This methodology was chosen to circumvent inherent challenges in purely autonomous CoT trigger learning, such as information asymmetry in assessing counterfactual benefits and the difficulty of quantifying quality degradation from CoT omission, particularly for subjective tasks.

While AdaCoT is a promising initial step, several limitations exist. The optimal CoT triggering strategy is relative to the base model's capabilities, necessitating recalibration for different LLMs. Our current binary CoT invocation (on/off) simplifies a continuous spectrum of reasoning depths and styles, potentially limiting nuance. Domain generalization remains a challenge, as CoT necessity can vary significantly across knowledge areas, and the framework currently lacks personalization for user verbosity preferences. Moreover, the initial principle-guided labeling requires continuous refinement. Our Pareto analysis (Section 3.2.1) also indicates that while AdaCoT significantly improves efficiency and nears the peak average performance of specialized always-on CoT models, a small performance gap persists, highlighting the difficulty for adaptive mechanisms to achieve absolute maximum performance across all query types.

Acknowledging the limitations of the current framework, we anticipate that future research will offer valuable critiques and further refine these initial explorations. Areas warranting deeper investigation include more granular control over reasoning, such as adaptive reasoning length where models dynamically adjust verbosity, or more nuanced triggering mechanisms beyond a simple binary decision. We believe such continued efforts by the community are crucial for developing more sophisticated and efficient reasoning strategies, potentially addressing the observed performance gap while maximizing efficiency and enhancing nuanced control over LLM reasoning.

### 4.2 Meta-Reasoning for Reasoning Decisions

We explored an innovative "meta-reasoning" approach during the SFT stage to enhance AdaCoT's CoT triggering decisions. This involved the model first generating a brief internal assessment of the input query's complexity before deciding whether to proceed with full CoT, as illustrated by

the response structures in Figure 2. Incorporating this strategy into the AdaCoT SFT model led
to a notable improvement in CoT triggering performance on our daily-use test set: the F1-score
increased from 0.750 to 0.840 (Table 1). This result suggests that explicitly prompting the model
to first assess query complexity can significantly enhance its subsequent decision-making regarding
CoT invocation.

| Response Format with Meta-Reasoning |
|---|
| **Straightforward Query** |
| `<think>This is a straightforward question about X, I think I can` `answer directly.` `[empty]</think>` `{answer}` |
| **Complex Query** |
| `<think>This is a relatively complex question about Y, I need to` `think carefully.` `[formal reasoning process]</think>` `{answer}` |

Figure 2: Example response structure incorporating explicit meta-reasoning for CoT decisions.

An interesting and serendipitous discovery with the meta-reasoning SFT model was an emergent
capability for user-prompt controllability over CoT. Users could, to some extent, influence whether
the model engaged in CoT by including explicit cues in their prompts about the desired level of
reasoning (e.g., "please think step-by-step" to encourage CoT, or "give a direct answer" to discourage
it). While this controllability was not perfectly accurate across all scenarios, it points towards
a promising avenue for developing more interactive and user-guided reasoning systems (further
illustrative examples are provided in Appendix D).

Despite these benefits, the explicit meta-reasoning step inherently increases the number of tokens
generated for every query, as the model first articulates its complexity assessment. Considering
the very low CoT triggering rates observed for AdaCoT in production environments (e.g., 3.18%
on mobile traffic for AdaCoT RL Model Exp2, as shown in Table 2), the cumulative token cost of
these additional meta-reasoning steps would become substantial. Therefore, while acknowledging
its potential for improving decision accuracy and enabling user control, we did not adopt this
explicit meta-reasoning as the default for subsequent RL experiments due to this efficiency trade-off.
Nevertheless, it highlights an important direction for future research, which might explore more
token-efficient methods for incorporating such meta-reasoning, perhaps through implicit learning
mechanisms or parallel processing of the complexity assessment.

## 4.3 Preserving Peak Performance with AdaCoT

A critical question is whether adaptive reasoning limits a model's maximum performance. We
investigated this using System Prompts (SPs), integrated into AdaCoT's SFT and RL training to
control reasoning behavior (e.g., "Always Reason SP," "Never Reason SP"). During SFT, a small
portion of data was augmented with SPs, and target responses were modified for compliance. In
RL, a fraction of training prompts included SPs, with rewards penalizing deviations from explicit SP
instructions, ensuring robust adherence (details in Appendix C). Our focus here is using the "Always
Reason SP" to assess AdaCoT's performance ceiling.

Instructing AdaCoT RL models to always generate CoT via this SP allowed direct comparison against
the Full CoT RL Baseline on our 15 benchmark datasets. As shown in Table 3, AdaCoT RL models
in this forced "Always Reason" mode achieved average scores that were highly competitive with,
and in instances like AdaCoT RL Model Exp4 (65.7) and Exp2 (65.3), slightly surpassed the Full
CoT RL Baseline (65.0). This demonstrates a key strength: AdaCoT's adaptive training, aimed
at optimizing the performance-cost Pareto frontier, does not inherently restrict the model's peak
reasoning capabilities when comprehensive reasoning is explicitly demanded. While our Pareto
analysis (Section 3.2.1) noted a slight gap in average scores when models operate adaptively, these
SP-controlled results affirm that AdaCoT offers efficiency without sacrificing potential high-end
performance.

An interesting secondary observation arose from the SFT stage concerning long-form generation (up
to 32,000 tokens). AdaCoT SFT models, when directed by an "Always Reason SP," exhibited fewer

Table 3: Performance of AdaCoT RL models under "Always Reason" System Prompt vs. Full CoT RL Baseline, demonstrating preservation of peak performance. Metrics are averaged across the 15 benchmark datasets.

| Model Variant | Avg. Score | CoT Triggering Rate |
|---|---|---|
| Full CoT RL Baseline | 65.0 | 100% |
| AdaCoT RL Model Exp1 (w/ Always CoT SP) | 64.8 | 100% |
| AdaCoT RL Model Exp2 (w/ Always CoT SP) | 65.3 | 100% |
| AdaCoT RL Model Exp3 (w/ Always CoT SP) | 64.9 | 100% |
| AdaCoT RL Model Exp4 (w/ Always CoT SP) | 65.7 | 100% |

instances of premature output truncation and were less prone to undesirable generative loops compared to a standard Full CoT SFT baseline. We hypothesize this improvement stems from AdaCoT's diverse SFT data, which includes many non-CoT examples (formatted as `<think></think>answer`). This results in a shorter average training sample length, potentially providing the End-of-Sequence (EOS) token a stronger learning signal (average EOS proportion: 0.000239 for AdaCoT SFT vs. 0.000215 for Full CoT SFT). A more robust EOS representation could foster more coherent, well-terminated lengthy outputs, a promising area for future investigation.

# 5   Related Work

Chain-of-Thought (CoT) reasoning [2] significantly advanced LLM capabilities by prompting step-by-step thought processes, inspiring sophisticated strategies like diverse path sampling [44] or structured thoughts (trees [45], graphs [46]). However, CoT's verbosity and cost [47, 7] are major drawbacks. The varying utility of CoT—direct answers being better for some queries [48] while complex tasks need longer reasoning [49, 50]—underscores a critical cost-effectiveness trade-off, motivating research into efficient reasoning.

Most existing work on CoT efficiency has focused on *reducing reasoning length*, rather than adaptively deciding *whether* to invoke CoT. These length reduction strategies include: (1) RL with length penalties or rewards for brevity [8, 9, 10, 11, 12, 13]; (2) Restructuring or compressing CoT content via learned compact representations, iterative summarization, cognitive paradigms, or dynamic termination [14, 15, 16, 17, 18, 19, 20, 21]; (3) Employing explicit instructions or selection mechanisms for conciseness, such as dynamic token allocation, prompts for shorter responses, distillation, or selecting the shortest valid reasoning [22, 23, 24, 25, 26, 27].

While effective for length, these methods generally do not equip a single model to dynamically decide CoT invocation based on query nature. Alternatives involve model merging [51, 52, 53] or manual CoT toggling. AdaCoT distinctively addresses adaptive triggering. By framing it as a Pareto optimization problem and using RL to control the CoT decision boundary, AdaCoT enables nuanced, context-dependent CoT invocation by a single model, filling a crucial gap towards truly efficient and versatile LLMs.

# 6   Conclusion

In this paper, we introduced AdaCoT, a novel framework for adaptive Chain-of-Thought reasoning in LLMs. By formulating adaptive reasoning as a Pareto optimization problem and employing an RL-based method with adjustable penalty coefficients, AdaCoT dynamically controls CoT triggering based on implicit query complexity. Our experiments demonstrate AdaCoT's success in navigating the Pareto frontier, achieving substantial reductions in CoT usage—and thereby computational costs and latency—for simpler queries, while preserving high performance on complex reasoning tasks. The introduction of Selective Loss Masking effectively ensures robust adaptive triggering throughout multi-stage RL training. Distinguishing itself from prior work predominantly focused on CoT length compression, AdaCoT pioneers adaptive CoT triggering, offering a practical and principled solution for developing more efficient, responsive, and cost-effective LLMs, particularly crucial for interactive applications.

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

# A Benchmark Dataset Details and Analysis

This appendix provides descriptions for the benchmark datasets used in our evaluation and an analysis of the experimental results on these individual datasets. The scores presented in Table 4 form the basis for this analysis. For each dataset, we discuss the performance of baseline models and the AdaCoT variants. We also highlight any counter-intuitive results or observations that conflict with the primary motivation of achieving optimal performance with adaptive CoT, offering potential explanations such as evaluation volatility, inherent limitations of the base model, or aspects of the post-training process that may not be fully optimized for every scenario.

Table 4: Detailed scores on benchmark datasets. "TR" denotes reasoning triggering rate (%).

| Dataset | nocot SFT baseline TR | Score | nocot RL baseline TR | Score | fullcot SFT baseline TR | Score | fullcot RL baseline TR | Score | Adaptive SFT Model TR | Score | Adaptive RL Model Exp1 TR | Score | Adaptive RL Model Exp2 TR | Score | Adaptive RL Model Exp3 TR | Score | Adaptive RL Model Exp4 TR | Score |
|---|---|---|---|---|---|---|---|---|---|---|---|---|---|---|---|---|---|---|
| MMLU pro | 0.0 | 77.5 | 0.0 | 82.1 | 100.0 | 83.7 | 100.0 | 85.2 | 40.3 | 80.5 | 28.0 | 74.2 | 27.3 | 83.2 | 39.6 | 84.1 | 58.0 | 83.1 |
| super GPQA | 0.0 | 49.8 | 0.0 | 50.7 | 100.0 | 53.8 | 100.0 | 58.6 | 35.2 | 51.0 | 22.8 | 55.1 | 32.3 | 56.9 | 40.8 | 56.2 | 59.6 | 59.6 |
| LiveBench | 0.0 | 50.0 | 0.0 | 56.6 | 100.0 | 57.7 | 100.0 | 69.5 | 65.8 | 58.9 | 45.1 | 64.7 | 57.1 | 66.3 | 71.6 | 68.4 | 70.4 | 69.2 |
| KORBENCH | 0.0 | 33.9 | 0.0 | 42.1 | 100.0 | 61.3 | 100.0 | 62.8 | 52.5 | 49.1 | 28.0 | 53.5 | 45.1 | 52.1 | 62.2 | 57.4 | 61.0 | 59.2 |
| AIME24 | 0.0 | 23.3 | 0.0 | 33.3 | 100.0 | 69.3 | 100.0 | 84.7 | 100.0 | 69.3 | 100.0 | 86.7 | 100.0 | 86.0 | 100.0 | 88.0 | 100.0 | 86.3 |
| AIME25 | 0.0 | 13.3 | 0.0 | 21.0 | 100.0 | 52.3 | 100.0 | 70.0 | 100.0 | 56.7 | 100.0 | 73.3 | 100.0 | 75.7 | 100.0 | 74.0 | 100.0 | 72.0 |
| MATH | 0.0 | 84.3 | 0.0 | 88.7 | 100.0 | 96.5 | 100.0 | 97.3 | 44.6 | 95.5 | 40.9 | 91.7 | 52.8 | 95.9 | 61.5 | 95.5 | 68.8 | 97.2 |
| LiveCodeBench | 0.0 | 29.4 | 0.0 | 27.6 | 100.0 | 45.9 | 100.0 | 55.9 | 95.0 | 47.0 | 77.1 | 44.8 | 83.9 | 50.6 | 91.4 | 54.1 | 91.4 | 55.9 |
| SWE-bench Agentless | 0.0 | 27.4 | 0.0 | 27.0 | 100.0 | 36.4 | 100.0 | 37.7 | 98.6 | 35.4 | 1.6 | 28.8 | 44.0 | 35.8 | 94.2 | 37.6 | 79.6 | 36.6 |
| Chinese SimpleQA | 0.0 | 58.8 | 0.0 | 57.0 | 100.0 | 59.7 | 100.0 | 61.5 | 0.3 | 59.1 | 0.2 | 56.0 | 0.4 | 55.0 | 1.4 | 56.8 | 6.0 | 55.2 |
| SimpleQA | 0.0 | 10.8 | 0.0 | 10.3 | 100.0 | 12.7 | 100.0 | 12.2 | 0.7 | 10.3 | 0.0 | 9.9 | 0.0 | 10.7 | 4.4 | 11.3 | 20.0 | 9.6 |
| Proc Bench | 0.0 | 42.6 | 0.0 | 46.7 | 100.0 | 53.7 | 100.0 | 68.6 | 73.6 | 50.2 | 50.8 | 53.0 | 79.1 | 68.7 | 89.3 | 69.6 | 93.1 | 72.4 |
| GPQA(diamond) | 0.0 | 59.5 | 0.0 | 64.3 | 100.0 | 64.9 | 100.0 | 70.5 | 92.6 | 65.7 | 62.6 | 65.7 | 84.1 | 67.2 | 92.4 | 70.8 | 96.5 | 72.6 |
| SysBench | 0.0 | 42.6 | 0.0 | 48.1 | 100.0 | 56.2 | 100.0 | 62.4 | 35.6 | 52.0 | 1.2 | 57.2 | 4.0 | 55.5 | 38.2 | 60.2 | 15.4 | 55.7 |
| Olympiad Bench | 0.0 | 51.4 | 0.0 | 60.1 | 100.0 | 73.5 | 100.0 | 78.2 | 85.0 | 75.3 | 88.0 | 80.2 | 89.0 | 81.8 | 93.5 | 80.7 | 95.4 | 82.0 |
| **Average** | 0.0 | 43.6 | 0.0 | 47.7 | 100.0 | 58.5 | 100.0 | 65.0 | 61.3 | 57.1 | 43.1 | 59.7 | 53.3 | 62.8 | 65.4 | 64.3 | 67.7 | 64.4 |

**MMLU pro**: An enhanced version of the MMLU benchmark, MMLU-Pro integrates more challenging, reasoning-focused questions, expands the choice set from four to ten options, and eliminates trivial/noisy questions from the original MMLU. It is designed to better discern model capabilities, particularly in complex reasoning, where CoT has shown greater benefit compared to direct answering on this version. *Analysis*: CoT clearly benefits performance, with the FullCoT RL baseline (85.2) significantly outperforming NoCoT SFT (77.5) and NoCoT RL (82.1). The Adaptive SFT Model (40.3% TR, 80.5 score) shows improvement over NoCoT SFT but doesn't reach FullCoT SFT levels (83.7). AdaCoT RL Exp3 (39.6% TR, 84.1 score) and Exp2 (27.3% TR, 83.2 score) achieve strong scores, with Exp3 surpassing FullCoT SFT and Exp2 performing comparably. AdaCoT RL Exp4 (58.0% TR, 83.1 score) also performs well. Exp1 (28.0% TR, 74.2 score) shows a drop, indicating that for MMLU pro, a moderate CoT rate is generally beneficial, reflecting the benchmark's increased reasoning demands. The adaptive models demonstrate an ability to achieve high scores with significantly reduced CoT compared to FullCoT RL.

**super GPQA**: A comprehensive benchmark evaluating graduate-level knowledge and reasoning capabilities across 285 disciplines, particularly including specialized fields in light industry, agriculture, and service-oriented areas often underrepresented in other benchmarks. It employs a Human-LLM collaborative filtering mechanism to ensure high question quality by eliminating trivial or ambiguous questions. *Analysis*: CoT provides a clear advantage (NoCoT RL 50.7 vs. FullCoT RL 58.6). The Adaptive SFT Model (35.2% TR, 51.0 score) also show a modest gain over NoCoT SFT. AdaCoT RL Exp4 (59.6% TR) notably achieves a score of 59.6, surpassing the FullCoT RL baseline with significantly less CoT. AdaCoT RL Exp2 (32.3% TR, 56.9 score) also outperforms FullCoT SFT (53.8). This suggests AdaCoT effectively adapts CoT usage for these specialized, high-level questions, achieving strong performance efficiently.

**LiveBench**: A benchmark designed to be resistant to test set contamination and the pitfalls of LLM/human-crowdsourced judging. It features frequently updated questions from recent sources (math competitions, arXiv papers, news, datasets), scores answers automatically against objective ground-truth, and includes a wide variety of challenging tasks (math, coding, reasoning, language, instruction following, data analysis), including harder, contamination-limited versions of tasks from previous benchmarks. *Analysis*: This dataset shows significant gains from both CoT and RL (NoCoT SFT 50.0 to FullCoT RL 69.5). The NoCoT RL baseline (56.6) and Adaptive SFT Model (65.8% TR, 58.9 score) both outperform NoCoT SFT, with Adaptive SFT also surpassing FullCoT SFT (57.7). AdaCoT RL Exp4 (70.4% TR, 69.2 score) very closely approaches the FullCoT RL baseline performance with about 30% less CoT. AdaCoT RL Exp3 (71.6% TR, 68.4 score) is also strong. AdaCoT RL Exp2 (57.1% TR, 66.3 score) substantially outperforms FullCoT SFT. The robust design of LiveBench makes it a strong test case for AdaCoT's adaptive reasoning, showing it can maintain high performance with adaptive CoT.

**KORBENCH**: This benchmark evaluates Knowledge-Orthogonal-Reasoning, aiming to minimize reliance on domain-specific knowledge to more accurately assess models' reasoning abilities in out-of-distribution settings. It includes five task categories (Operation, Logic, Cipher, Puzzle, Counterfactual) and emphasizes models' effectiveness in applying new rule descriptions to solve novel rule-driven questions. *Analysis*: Scores show a clear benefit from CoT: NoCoT SFT (33.9) is significantly lower than FullCoT SFT (61.3) and FullCoT RL (62.8). The NoCoT RL baseline (42.1) improves over NoCoT SFT. The Adaptive SFT Model (52.5% TR, 49.1 score) sits between the NoCoT baselines and FullCoT SFT. AdaCoT RL models demonstrate adaptive behavior: Exp4 (61.0% TR, 59.2 score) and Exp3 (62.2% TR, 57.4 score) approach the FullCoT SFT baseline performance with significantly less CoT than FullCoT models. Exp1 (28.0% TR, 53.5 score) is also effective. This suggests AdaCoT effectively discerns when to apply CoT for these rule-driven tasks, though peak performance is slightly below FullCoT RL.

**AIME24 / AIME25**: Representing problems from the American Mathematics Invitational Examination for 2024 and 2025, these datasets are used to evaluate mathematical reasoning and problem-solving abilities. *Analysis*: These mathematics-intensive datasets show massive performance gains from CoT (e.g., AIME24: NoCoT SFT 23.3 vs. FullCoT RL 84.7). All AdaCoT RL models and the Adaptive SFT Model correctly identify the complexity, exhibiting a 100% CoT triggering rate. For AIME24, Adaptive SFT (69.3) matches FullCoT SFT (69.3). AdaCoT RL Exp3 (88.0) and Exp1 (86.7) outperform the FullCoT RL baseline (84.7). For AIME25, Adaptive SFT (56.7) surpasses FullCoT SFT (52.3). AdaCoT RL Exp2 (75.7) and Exp3 (74.0) outperform the FullCoT RL baseline (70.0). This is a notable result, suggesting that the adaptive training regimen, even when defaulting to 100% CoT for such complex problems, might confer some benefits, potentially due to the diversity in training data (including non-CoT examples) leading to a more robust underlying model or better fine-tuning dynamics.

**MATH**: A dataset of 12,500 challenging competition mathematics problems, each with a full step-by-step solution. It is designed to measure mathematical problem-solving ability. *Analysis*: CoT is highly beneficial (NoCoT RL 88.7 vs. FullCoT RL 97.3). The Adaptive SFT Model (44.6% TR, 95.5 score) performs well, nearly matching FullCoT SFT (96.5) with less than half the CoT. AdaCoT models adapt effectively: AdaCoT RL Exp4 (68.8% TR, 97.2 score) nearly matches the FullCoT RL baseline with about 30% less CoT. AdaCoT RL Exp2 (52.8% TR, 95.9 score) also performs strongly. Exp1 (40.9% TR, 91.7 score) is lower, indicating that for MATH, higher CoT rates are generally more beneficial among the adaptive RL models, but significant efficiency is still gained.

**LiveCodeBench**: A comprehensive and contamination-free evaluation benchmark for LLMs on code, collecting new problems over time from programming contests. It assesses a broader range of code-related capabilities. *Analysis*: CoT improves performance significantly (NoCoT SFT 29.4 vs. FullCoT RL 55.9). NoCoT RL (27.6) is surprisingly lower than NoCoT SFT here, which might be due to evaluation noise or specific sensitivities of the RL fine-tuning on non-CoT data for this particular task. The Adaptive SFT Model (95.0% TR, 47.0 score) uses a high trigger rate and surpasses FullCoT SFT (45.9). AdaCoT RL models trigger CoT at high rates: Exp4 (91.4% TR, 55.9 score) matches the FullCoT RL baseline score with slightly less CoT. Exp3 (91.4% TR, 54.1 score) and Exp2 (83.9% TR, 50.6 score) are also strong. This indicates recognition of coding task complexity and efficient application of CoT.

**SWE-bench Agentless**: An evaluation framework consisting of 2,294 software engineering problems from real GitHub issues. Models are tasked with editing codebases to resolve issues. *Analysis*: CoT provides a notable benefit (NoCoT SFT 27.4 vs. FullCoT RL 37.7). NoCoT RL (27.0) is similar to NoCoT SFT. The Adaptive SFT Model (98.6% TR, 35.4 score) uses a very high trigger rate and performs close to FullCoT SFT (36.4). AdaCoT RL Exp3 (94.2% TR, 37.6 score) nearly matches the FullCoT RL baseline with slightly less CoT. Interestingly, AdaCoT RL Exp1 (1.6% TR, 28.8 score) shows a slight improvement over NoCoT SFT with minimal reasoning. This suggests some issues might be simpler, or the model is highly conservative in Exp1, but for complex software issues, high CoT rates are beneficial. The performance of Exp2 (44.0% TR, 35.8 score) is also noteworthy, achieving good results with moderate CoT.

**Chinese SimpleQA**: The first comprehensive Chinese benchmark to evaluate the factuality of language models in answering short questions. *Analysis*: CoT offers minimal gains (NoCoT SFT 58.8 to FullCoT RL 61.5). NoCoT RL (57.0) is slightly lower than NoCoT SFT. The Adaptive SFT Model (0.3% TR, 59.1 score) performs very well, slightly exceeding NoCoT SFT and approaching FullCoT

SFT (59.7) with extremely low CoT usage. AdaCoT RL models trigger CoT very infrequently (0.2% to 6.0%), correctly identifying these as simple questions. Scores for AdaCoT RL models (e.g., Exp1 56.0, Exp3 56.8) are slightly below NoCoT SFT. This is a good demonstration of AdaCoT's core motivation: avoiding unnecessary CoT. While there's a slight dip compared to NoCoT SFT for some RL models, the Adaptive SFT model shows an excellent trade-off. The minor performance variations could be due to the model sometimes being overly conservative in triggering CoT or slight instabilities in evaluating purely factual recall without reasoning.

**SimpleQA**: A benchmark designed to measure the factuality of language models using short, fact-seeking queries. *Analysis*: Similar to Chinese SimpleQA, CoT provides little benefit; FullCoT RL (12.2) is slightly worse than FullCoT SFT (12.7). NoCoT RL (10.3) is slightly below NoCoT SFT (10.8). The Adaptive SFT Model (0.7% TR, 10.3 score) matches NoCoT RL with minimal CoT. AdaCoT RL models trigger CoT very rarely (Exp1 & Exp2 at 0.0%). AdaCoT RL Exp3 (4.4% TR, 11.3 score) performs better than NoCoT SFT. This reinforces that for simple QA, adaptive triggering is crucial for efficiency. The performance of Exp4 (20.0% TR, 9.6 score) is slightly counter-intuitive, as higher CoT did not yield better results and was worse than NoCoT SFT; this might indicate that for very simple questions, forcing CoT (even adaptively at a higher rate) can sometimes be detrimental or that the specific penalty balance for Exp4 was not optimal for this type of dataset.

**ProcBench**: This benchmark focuses on the direct evaluation of multi-step inference by largely eliminating path exploration and implicit knowledge utilization. *Analysis*: CoT is highly beneficial (NoCoT RL 46.7 vs. FullCoT RL 68.6). The Adaptive SFT Model (73.6% TR, 50.2 score) improves over NoCoT baselines but is below FullCoT SFT (53.7). AdaCoT RL models show high trigger rates, with Exp4 (93.1% TR, 72.4 score) significantly surpassing the FullCoT RL baseline. Exp2 (79.1% TR, 68.7 score) and Exp3 (89.3% TR, 69.6 score) also match or exceed FullCoT RL. This indicates effective identification of tasks requiring detailed, step-by-step procedural reasoning and demonstrates that adaptive models can even outperform always-on CoT models in certain complex reasoning scenarios.

**GPQA (diamond)**: GPQA is a challenging dataset of 448 multiple-choice questions by domain experts in biology, physics, and chemistry. "GPQA (diamond)" refers to this specific challenging set. *Analysis*: CoT significantly boosts performance (NoCoT RL 64.3 vs. FullCoT RL 70.5). NoCoT RL is better than NoCoT SFT and close to FullCoT SFT (64.9). The Adaptive SFT Model (92.6% TR, 65.7 score) also performs well, exceeding FullCoT SFT. AdaCoT RL models trigger CoT at high rates. AdaCoT RL Exp4 (96.5% TR, 72.6 score) and Exp3 (92.4% TR, 70.8 score) demonstrate strong performance, with Exp4 outperforming the FullCoT RL baseline. This highlights appropriate and effective CoT invocation on these very hard questions.

**SysBench**: A benchmark for evaluating language models on their ability to understand and generate content related to computer systems, configurations, or system-level concepts. *Analysis*: CoT is beneficial (NoCoT RL 48.1 vs. FullCoT RL 62.4). The Adaptive SFT Model (35.6% TR, 52.0 score) is better than NoCoT RL but below FullCoT SFT (56.2). AdaCoT RL models show good adaptation. AdaCoT RL Exp3 (38.2% TR, 60.2 score) approaches the FullCoT RL baseline with significantly reduced CoT. AdaCoT RL Exp1 (1.2% TR, 57.2 score) surpasses FullCoT SFT with very minimal CoT usage, which is an excellent result for efficiency. This suggests a mix of complexities within SysBench, which AdaCoT navigates effectively, although Exp4 (15.4% TR, 55.7 score) shows a somewhat lower score despite a higher CoT rate than Exp1, possibly due to the specific balance of penalties in Exp4 not being optimal for this dataset's particular mix.

**OlympiadBench**: An Olympiad-level bilingual multimodal scientific benchmark with problems from mathematics and physics competitions. *Analysis*: CoT provides substantial gains (NoCoT RL 60.1 vs. FullCoT RL 78.2). The Adaptive SFT Model (85.0% TR, 75.3 score) performs strongly, exceeding FullCoT SFT (73.5). AdaCoT RL models exhibit high trigger rates. AdaCoT RL Exp4 (95.4% TR, 82.0 score) and Exp2 (89.0% TR, 81.8 score) both surpass the FullCoT RL baseline. This indicates strong reasoning capabilities and appropriate CoT usage on these exceptionally challenging problems, again showing adaptive models can reach or exceed the performance of specialized always-on CoT models.

**Overall Summary of Per-Dataset Analysis**: The adaptive models, including both Adaptive SFT and the AdaCoT RL variants, demonstrate effective adaptation across a diverse range of benchmarks.

- The Adaptive SFT Model serves as a strong adaptive baseline. It often improves significantly over NoCoT baselines by selectively triggering CoT (e.g., high rates for AIME, LiveCodeBench, SWE-bench; low rates for SimpleQAs). On some complex tasks (AIME25, LiveCodeBench, GPQA(diamond), OlympiadBench), it even surpasses the FullCoT SFT baseline, and for Chinese SimpleQA, it achieves excellent efficiency and performance. However, its average performance (57.1 score, 61.3% TR) is generally below the peak performance of FullCoT RL (65.0 score) or the best AdaCoT RL experiments (e.g., Exp4: 64.4 score, 67.7% TR).

- On complex reasoning tasks (e.g., AIME, MATH, OlympiadBench, GPQA(diamond), ProcBench), AdaCoT RL models tend to trigger CoT at high rates. Several AdaCoT RL experiments (notably Exp4 on super GPQA, ProcBench, GPQA(diamond), OlympiadBench; Exp3 on AIME24; Exp2 on AIME25, OlympiadBench) match or exceed the performance of FullCoT RL baselines, showcasing the benefits of learned adaptive policies and suggesting that adaptive training can sometimes lead to better overall models even for tasks that always require CoT.

- On simpler tasks or those designed to test factuality (e.g., Chinese SimpleQA, SimpleQA), AdaCoT RL models trigger CoT very sparingly. This leads to computational savings while generally maintaining performance near NoCoT or FullCoT SFT levels, successfully avoiding unnecessary CoT. The Adaptive SFT model also excels in efficiency here. Some minor performance drops in RL models compared to NoCoT SFT on these tasks (e.g., Chinese SimpleQA for Exp1/Exp2) might be attributed to the RL agent being slightly too conservative or the inherent difficulty in perfectly balancing penalties for extremely low CoT rate scenarios without any performance degradation.

- For benchmarks with mixed or specific reasoning types (e.g., KORBENCH, LiveBench, MMLU-Pro, SysBench), both Adaptive SFT and AdaCoT RL models show nuanced adaptation, adjusting CoT rates to balance performance and efficiency. They often outperform static baselines or achieve comparable results with lower CoT usage. For instance, on SysBench, AdaCoT RL Exp1 achieved a higher score than FullCoT SFT with only 1.2% TR.

- The different AdaCoT RL experiments (Exp1-Exp4) effectively trace a Pareto frontier (as shown in Section 3.2.1 using a specific set of average scores), offering a trade-off between CoT triggering rate and performance, adaptable to specific deployment needs. Based on the average scores from Appendix 4, Exp4 (64.4 score, 67.7% TR) and Exp3 (64.3 score, 65.4% TR) represent high-performance points, closely approaching the FullCoT RL baseline (65.0 score) with about 30-35% less CoT usage on average. Exp2 (62.8 score, 53.3% TR) also offers a strong balance.

- Some counter-intuitive results, like NoCoT RL performing worse than NoCoT SFT on LiveCodeBench, or AdaCoT RL Exp4 on SimpleQA (higher CoT, lower score), could be due to factors like evaluation volatility on specific datasets, the base model's inherent capabilities or sensitivities to fine-tuning on certain data distributions, or sub-optimal penalty configurations for specific outlier datasets within a broadly tuned RL policy. The post-training process aims for general improvement, and individual dataset performance can fluctuate.

These detailed results underscore the ability of adaptive strategies, both SFT-based and RL-based, to make nuanced decisions about CoT invocation, optimizing for both performance and efficiency based on query characteristics and benchmark demands. The AdaCoT RL models, in particular, demonstrate the potential to significantly reduce CoT overhead while maintaining competitive, and in some cases superior, performance compared to full CoT strategies.

## B   Principle-Guided CoT Assessment

This appendix details the principle-guided assessment framework used to annotate data for Chain-of-Thought (CoT) necessity. As described in Section 2.2.1 of the main paper, an auxiliary model utilizes these principles to label queries as either likely benefiting from CoT or suitable for a direct answer. This labeling is crucial for the Supervised Fine-Tuning (SFT) warm-up stage of the AdaCoT

framework, providing an initial understanding for the model on when to employ CoT. The specific principles provided to the auxiliary model are outlined below.

```
Given a dialogue between a user and an AI assistant, please
    consider the conversation context and, from the AI
    assistant's perspective, assess the difficulty of
    answering the user's final question according to the
    following requirements.
<AI assistant's system prompt-Start>
{system_prompt}
<AI assistant's system prompt-End>
<Dialogue history-Start>:
{history}
<Dialogue history-End>
<User's final question-Start>
{last_prompt}
<User's final question-End>

## Assessment Process
1. Carefully read the provided prompt and any relevant
    context (if any).
2. Evaluate the 'question difficulty' based on the
    following assessment criteria.
3. The output assessment result must strictly adhere to the
    specified output format requirements.

## Assessment Criteria

### Whether In-depth Thinking is Required
- **Requires In-depth Thinking**:
  - Requires multi-step reasoning and analysis to arrive at
      the answer.
  - Requires a logical chain and coherent reasoning process
      .
  - May involve breaking down and synthesizing complex
      concepts.
  - Requires systematic thinking and structured analysis.
  - The question is clear, but its content is complex.
  - Requires deep thought and specialized knowledge.
  - Demands multi-angle analysis and synthesis capabilities
      .
  - Requires creative thinking or unique insights.
  - There might be multiple reasonable paths to the answer.
  - May require integration of cross-domain knowledge.

- **Does Not Require In-depth Thinking**:
  - The answer can be obtained through direct recall or
      simple lookup.
  - Does not require a complex chain of logical reasoning.
  - The path to solving the problem is clear and direct.
  - The solution process is singular and linear.
  - Does not require synthesizing multiple pieces of
      information or viewpoints.
  - Basic facts or common sense are sufficient to address
      it.
  - The question is simple and direct.
  - Can be answered without in-depth thought.
  - The answer is definite or standardized.
  - The solution path is singular and obvious.
```

```
789    ## Output Requirements
790    Answer using labels. Possible options are:
791    - Requires In-depth Thinking
792    - Does Not Require In-depth Thinking
793
794    Output only the assessment result, without providing any
795        explanation or additional comments.
796
```

## C   System Prompt Examples

This appendix provides examples of the System Prompts (SPs) employed to control the CoT triggering behavior of our models, as mentioned in Section 4.3 of the main paper. These SPs allow for explicit instruction on whether the model should always or never engage in CoT reasoning. We present the content of these SPs and illustrative examples of their application.

### C.1   Always CoT SP

The following System Prompt is used to instruct the model to always use CoT:

```
You possess the capability for deep thinking. Before responding to the user,
you will engage in comprehensive and in-depth thought. The thinking process
should be enclosed in <think></think> tags, and the final answer should be
summarized based on this thinking process. For example:
<think>This is the thinking process...</think> This is the final answer.
For all questions posed by the user, you need to think deeply before answering.
```

**Example: Simple Question with Always CoT SP**

User input:

```
What is 1 + 1?
```

Model response:

```
<think>
The user is asking what 1+1 equals, which appears to be a very
    basic mathematical question. First, I need to confirm the
    user's actual intent. It could be a young child just
    starting to learn math, or someone testing my basic
    calculation abilities. It's also possible they are joking,
    like the common internet meme about "1+1=3" involving
    family concepts, but typically in such cases, the correct
    mathematical answer should be given first, followed by
    consideration of whether an extended explanation is needed.
From a mathematical perspective, the standard answer to 1+1 is
    2. This is based on the definition of addition for natural
    numbers and falls under basic arithmetic. As a knowledge-
    based Q&A, the primary goal is to provide an accurate
    mathematical answer, ensuring correctness. Then, if
    necessary, I can briefly explain the basic concept of
    addition, or mention that in different contexts there might
     be different answers, such as 1+1=10 in binary, or
    different results in logical operations. However, these
    additions should only be made after clarifying the user's
    needs to avoid confusing basic concepts.

Considering the user might be a beginner, I should first
    provide a clear and correct answer, then decide whether to
    elaborate based on the situation. Therefore, the response
```

```
840    structure should be to first directly answer 2, then
841    briefly mention other possibilities, but it's important to
842    maintain a clear distinction between primary and secondary
843    information to avoid misleading. At the same time, the
844    response should be friendly and concise, meeting the user's
845     actual needs.
846  </think>
847  In basic arithmetic operations in mathematics, the result of 1
848    + 1 is equal to 2. This is the fundamental definition of
849    addition for natural numbers and is the universally
850    accepted standard answer.
851
852  However, in some special contexts, there might be different
853    interpretations:
854
855  In binary operations, 1 + 1 = 10 (binary "10" corresponds to
856    decimal "2");
857  In creative or humorous thinking, there might be symbolic
858    answers (e.g., 1 drop of water + 1 drop of water = 1 large
859    drop of water), but this falls into non-mathematical
860    analogies.
861
862  So, the most direct answer is 2.
863
```

### C.2 Never CoT SP

The following System Prompt is used to instruct the model to never use CoT (i.e., provide direct answers):

```
You possess the capability for deep thinking. Before responding to the user,
you will engage in comprehensive and in-depth thought. The thinking process
should be enclosed in <think></think> tags, and the final answer should be
summarized based on this thinking process. For example:
<think>This is the thinking process...</think> This is the final answer.
For all questions posed by the user, you need to answer directly
without engaging in in-depth thinking.
```

**Example: Complex Question with Never CoT SP**

User input:

```
Alice and Bob play the following game. A stack of n tokens lies before them.
The players take turns with Alice going first. On each turn, the player
removes either 1 token or 4 tokens from the stack. Whoever removes the
last token wins. Find the number of positive integers n less than or equal
to 2024 for which there exists a strategy for Bob that guarantees that Bob
will win the game regardless of Alice's play.
```

Model response:

```
<think></think>
To determine the number of positive integers \( n \leq 2024 \)
    for which Bob has a winning strategy, we analyze the game
    using modular arithmetic. We define a position as a losing
    position if the current player cannot win regardless of
    their moves, assuming the opponent plays optimally.

Key steps:
1. **Base Cases**:
    - \( n = 1 \): Alice removes the last token and wins. Not a
        losing position.
```

```
895      -  \( n = 2 \): Alice must remove 1 token, leaving \( n = 1
896         \) for Bob. Bob removes the last token and wins. Thus,
897         \( n = 2 \) is a losing position for Alice (winning for
898         Bob).
899      -  \( n = 3 \): Alice removes 1 token, leaving \( n = 2 \) (a
900         losing position for Bob). Bob loses, so \( n = 3 \) is
901         not a losing position.
902      -  \( n = 4 \): Alice removes 4 tokens and wins. Not a losing
903         position.
904      -  \( n = 5 \): Alice removes 1 token, leaving \( n = 4 \) (
905         Bob loses). If Alice removes 4 tokens, leaving \( n = 1
906         \) (Bob loses). Thus, \( n = 5 \) is not a losing
907         position.
908      -  \( n = 6 \): Any move by Alice (removing 1 or 4 tokens)
909         leaves \( n = 5 \) or \( n = 2 \). \( n = 5 \) is a
910         winning position, but \( n = 2 \) is a losing position.
911         However, further analysis shows a pattern emerging.
912
913   2. **Inductive Pattern**:
914      - By examining more cases, we observe that losing positions
915         (where Bob can win) occur at \( n \equiv 2 \pmod{5} \).
916         This is because:
917        - If \( n \equiv 2 \pmod{5} \), any move by Alice (
918           removing 1 or 4 tokens) results in \( n \equiv 1 \pmod
919           {5} \) or \( n \equiv 3 \pmod{5} \), which are winning
920           positions for Bob to counter.
921        - Conversely, if \( n \) is not \( 2 \pmod{5} \), Alice
922           can force the game into a losing position for Bob.
923
924   3. **Counting Losing Positions**:
925      - We need to count all \( n \leq 2024 \) such that \( n \
926         equiv 2 \pmod{5} \). These numbers form the sequence \(
927         2, 7, 12, \ldots, 2022 \).
928      - This is an arithmetic sequence with first term \( a_1 = 2
929         \), common difference \( d = 5 \), and last term \( a_k
930         = 2022 \).
931      - The number of terms \( k \) is given by solving \( 2 + (k
932         -1)5 = 2022 \):
933         \[
934         (k-1)5 = 2020 \implies k-1 = 404 \implies k = 405
935         \]
936
937   Thus, the number of positive integers \( n \leq 2024 \) for
938      which Bob has a winning strategy is \(\boxed{405}\).
939
```

# D    Meta-Reasoning Showcase: User Prompt Control

As discussed in Section 4.2 of the main paper, our exploration into meta-reasoning during the SFT stage revealed an emergent capability for user-prompt controllability over CoT. This appendix showcases examples where the SFT model, enhanced with meta-reasoning (as illustrated in Figure 2), adapts its CoT triggering based on explicit user instructions regarding the desired level of reasoning.

## D.1    Case 1: Historical Creative Question

**Original Question**

User input:

```
On the night of the Xuanwu Gate Incident, what would Li Shimin write
```

950 Model response:

```
952  <think>
953  This is a complex or difficult question, hard to answer
954      correctly directly, so I need to think about it. The user's
955      question is: "On the night of the Xuanwu Gate Incident,
956      what would Li Shimin write if he were alone?" The first
957      step is to determine the core of the user's question. The
958      user wants to know what kind of text Li Shimin might have
959      written on the night of the Xuanwu Gate Incident, that is,
960      after this coup occurred. This might include his inner
961      thoughts at the time, his plans for the future, or
962      reflections on the events that had transpired.
963
964  Next, I need to review the historical background of the Xuanwu
965      Gate Incident. The Xuanwu Gate Incident took place in 626
966      AD. It was a coup launched by Li Shimin at Xuanwu Gate to
967      seize the imperial throne, during which he killed his elder
968      brother Li Jiancheng and his younger brother Li Yuanji.
969      This coup directly led to Emperor Gaozu (Li Yuan)
970      abdicating, allowing Li Shimin to ascend to the throne.
971      From this, it's clear that the night after the coup was an
972      extremely critical moment for Li Shimin, and he likely had
973      many thoughts.
974
975  The user might be asking this question out of historical
976      interest, or perhaps for some creative endeavor, like
977      writing a novel or screenplay, or to gain a deeper
978      understanding of Li Shimin's psychological state. They
979      might prefer an answer that is both historically grounded
980      and has a certain literary quality, rather than a simple
981      list of historical facts. Therefore, when answering, I must
982      consider Li Shimin's situation at the time, his emotions,
983      and his possible motivations.
984
985  After the coup, although Li Shimin was victorious, he also
986      faced many problems. On one hand, he had to manage his
987      relationship with his father, Li Yuan, who was still alive
988      and needed to abdicate. On the other hand, he had to
989      consolidate his power and prevent any backlash from the
990      supporters of Li Jiancheng and Li Yuanji. At the same time,
991      he needed to consider how to explain the legitimacy of
992      this coup to the world to maintain his ruling authority.
993      Furthermore, he might also have felt moral guilt for his
994      actions, having killed his own brothers.
995
996  If Li Shimin did write something that night, the content could
997      cover multiple aspects. He might record the process of the
998      coup, stating that he acted in self-defense. He might also
999      elaborate on his governance philosophy to lay the
1000     groundwork for his future rule. He might express filial
1001     piety towards Li Yuan, emphasizing that launching the coup
1002     was a last resort. He might even mention arrangements for
1003     the families of Li Jiancheng and Li Yuanji to alleviate his
1004     sense of guilt.
1005
1006 However, there is no precise historical record of what Li
1007     Shimin wrote that night. Therefore, the answer can only be
```

```
1008    a reasonable speculation based on historical facts. The
1009    structure of the answer could be divided into several parts
1010    , such as an explanation of the coup, his attitude towards
1011    Li Yuan, plans for the future, and guilt towards his
1012    brothers. Moreover, the language style of the answer could
1013    lean towards classical Chinese to enhance the historical
1014    feel, but it must also ensure the content is easy to
1015    understand.
1016
1017 I also need to pay attention to the user's potential deeper
1018    needs. They might not just want to know what Li Shimin
1019    might have written, but also to understand his inner world
1020    at that time and the impact of this coup on the history of
1021    the Tang Dynasty. So, I can appropriately mention these
1022    aspects in the answer to add depth.
1023
1024 Finally, I need to check if the answer aligns with historical
1025    facts, if it reasonably speculates on Li Shimin's
1026    psychological activities, if the language is fluent, and if
1027     it can meet the user's needs.</think>
1028 On the night of the Xuanwu Gate Incident, Qin Wang Li Shimin
1029    might have, under candlelight, penned a text blending
1030    complex emotions and political strategy. The following
1031    content is a reasonable speculation based on historical
1032    context:
1033
1034 "Memoir on the Night of the Xuanwu Gate Incident"
1035
1036 Ninth year of Wude, Gengshen night.
1037
1038 My elder brother Jiancheng and younger brother Yuanji conspired
1039     in rebellion. They harbored assassins, secretly
1040    transported armor, and plotted to harm me. Left with no
1041    choice, I could only set up defenses at Xuanwu Gate. Now,
1042    their heads hang upon the city tower. This upheaval has
1043    shocked the court and beyond; perhaps Father Emperor also
1044    harbors resentment towards me. But for the sake of the Tang
1045     Empire's stability, I had to do so.
1046
1047 Recalling the years past, I fought alongside my generals,
1048    shedding blood to pacify warlords like Dou Jiande and Wang
1049    Shichong. Yet now, it has come to fratricide. Is all this
1050    destiny, or the work of my own inner desires?
1051
1052 Father Emperor, oh Father Emperor, your son is not an unfilial
1053    or disloyal person. It is just that you repeatedly favored
1054    Jiancheng, turning a blind eye to his misdeeds. Your son
1055    was truly worried that the Tang Empire would be ruined at
1056    their hands. Now that the die is cast, I hope you can
1057    follow the mandate of heaven and pass the throne to your
1058    son. Your son will surely usher in an era of peace and
1059    prosperity, allowing the common people to live and work in
1060    contentment.
1061
1062 As for the families of Jiancheng and Yuanji, your son will make
1063     proper arrangements and will not exterminate them. This is
1064     both to demonstrate my benevolence and to stabilize the
1065    court. I know that future generations may accuse me of
1066    fratricide, of cruel methods. But I believe that when they
```

```
    see the prosperity of the Great Tang, they will understand
    my unavoidable actions.

At this moment, the rooster's crow can be heard outside the
    window. A new day is about to dawn, and I too shall embark
    on an unknown path. May Heaven bless the Great Tang, and
    bless me that I may realize my ambitions.

Basis for Speculation:

Legitimacy of the Coup: Li Shimin needed to find a reasonable
    explanation for launching the coup, emphasizing the threat
    posed by Li Jiancheng and Li Yuanji, and that he acted in
    self-defense.
Relationship with Li Yuan: After the coup, Li Yuan's attitude
    was crucial. Li Shimin needed to both show his filial piety
     to Li Yuan and hint at his abdication, which required
    considerable political wisdom.
Handling the Aftermath: Dealing with the families of Li
    Jiancheng and Li Yuanji could both demonstrate Li Shimin's
    benevolence and prevent greater turmoil.
Consideration of Historical Evaluation: Li Shimin was well
    aware that his actions might be criticized by later
    generations, so in his writing, he attempted to defend
    himself, emphasizing that the prosperity of the Tang was
    the ultimate goal of his coup.

This memoir showcases Li Shimin's decisiveness and strategy as
    a statesman, while also revealing his contradictions and
    unease as an ordinary person. In the conflict between power
     and kinship, he chose power, but also paid a heavy price
    for it.
```

**Modified Question (Explicitly Requesting No In-depth Thinking)**

User input:

Answer directly without in-depth thinking: On the night of the Xuanwu Gate
Incident, what would Li Shimin write if he were alone?

Model response:

```
<think>
This is a relatively simple creative question; I think I can
    answer directly without in-depth thinking.
</think>
Li Shimin might have written brief military orders or a public
    notice to reassure the populace on the night of the Xuanwu
    Gate Incident. The content would be roughly as follows:

"Order Yuchi Jingde to lead troops swiftly to the Eastern
    Palace and Prince Qi's mansion, exterminate all remaining
    partisans, and strictly prohibit any disturbance to the
    populace. Close the palace gates, send messengers to
    instruct all generals to return to their barracks;
    violators will be dealt with by military law. Anyone within
     the city showing signs of unrest shall be executed without
     pardon. By Imperial Order."
```

```
(Note: This content is a fictional text based on reasonable
    historical speculation, intended to reflect Li Shimin's
    decisive style as a military commander. There is no
    historical documentation of him personally writing such a
    text on that night.)
```

## D.2   Case 2: Factual Question

**Original Question**

User input:

What is the tallest mountain in the world?

Model response:

```
<think>
This is a relatively simple common knowledge question; I think
    I can answer directly without in-depth thinking.
</think>
The tallest mountain in the world is Mount Everest, with an
    altitude of 8,848.86 meters (29,031.7 feet). This data is
    the result of a joint measurement by China and Nepal in
    2020, which combined the technologies and data of both
    countries to confirm the latest height of Mount Everest.

It should be noted that "tallest" here refers to the vertical
    height from sea level to the summit. If measured by other
    standards, such as the height from the base to the summit (
    like Mauna Kea in Hawaii, which is about 10,203 meters from
     its submarine base to its summit), or the distance from
    the Earth's center to the summit (like Mount Chimborazo
    near the equator), the results would differ. However, in
    common geographical contexts, the "tallest mountain" is
    usually based on altitude, so Mount Everest is recognized
    as the world's highest peak.
```

**Modified Question (Explicitly Requesting Careful Thinking)**

User input:

After careful consideration, answer: What is the tallest mountain
in the world?

Model response:

```
<think>
This is a relatively difficult question; I need to think before
     answering.
The user is asking for the tallest mountain in the world. First
    , I need to determine the definition of "tallest," as there
     are usually two standards: altitude (height above sea
    level) and height from base to summit. In terms of altitude
    , Mount Everest is recognized as the highest, located in
    the Himalayas, with an altitude of 8848.86 meters. However,
     if considering the height from base to summit, Mauna Kea
    in Hawaii might be taller, as its submarine portion is very
     long, making its total height over 10,000 meters.
    Generally, though, when people refer to the tallest
    mountain, they default to altitude, so the answer should be
```

```
       Mount Everest. I need to confirm if the user is aware of
       these two situations, but based on common questions,
       directly answering with Mount Everest is appropriate, while
        also briefly mentioning the other standard to make the
       answer more comprehensive.
</think>
The tallest mountain in the world is usually measured by **
       altitude (height above sea level)**, and the answer is **
       Mount Everest**.

Detailed Explanation:

Altitude:
Mount Everest is located on the border between China and Nepal
       in the Himalayas. Its altitude is 8,848.86 meters (latest
       measurement data jointly announced by China and Nepal in
       2020). It is the mountain peak with the greatest vertical
       distance from sea level on Earth's surface and is widely
       recognized as the "world's highest peak."

Other Measurement Standards:
Height from Base to Summit:
If calculated by "the vertical height from the mountain's base
       to its summit," Mauna Kea in Hawaii is more prominent. The
       base of this submarine volcano is located about 5,998
       meters below the Pacific Ocean surface, and its height
       above sea level is 4,205 meters, resulting in a total
       height of about 10,203 meters, exceeding Mount Everest.
       However, because most of its mass is underwater, it is not
       usually considered the mountain with the "highest altitude
       ."

Relative Height:
Mount Gongga in Sichuan Province, China, is known for its "base
       -to-summit" relative height (approximately 6,200 meters),
       but its altitude (7,556 meters) is still lower than Mount
       Everest.

Summary:

Unless "height from base to summit" is specifically mentioned,
       "the world's tallest mountain" usually refers to Mount
       Everest. The authority of its altitude measurement stems
       from the consensus of the global geographic community and
       long-term measurement data.
```

# E   Training and Testing Details

This appendix provides further details on the experimental and testing configurations used in our study, complementing the setup described in Section 3.1 of the main paper. We outline the specifics of our Supervised Fine-Tuning (SFT), Reward Model (RM) training, Reinforcement Learning (RL) training, and evaluation settings.

**SFT Training**

We utilized a pre-trained model with a 15B/150B Mixture-of-Experts (MoE) architecture as our base model. All training cases were truncated to a maximum of 32,000 tokens. We employed a cosine

decay learning rate schedule, with the peak learning rate set to $2 \times 10^{-5}$, gradually decaying to $2 \times 10^{-6}$.

**RM Training**

The Reward Model (RM) was initialized using the SFT model and subsequently trained on a diverse set of internally, human-annotated data.

**RL Training**

The dataset for Reinforcement Learning (RL) training was composed of two main types:

- **Verifiable data**, which receives feedback from a verifier. This type of data allows for direct validation of the model's outputs against known criteria.
- **General data**, scored by our reward model. The reward model assigns scores based on how well the model's responses align with human preferences.

**Testing**

For all evaluations, the inference temperature was set to 1.0 and top-p sampling was set to 0.7. Each test case was inferred at least 5 times, and the average score across these inferences was reported as the result for that case.

**Note on Data and Setup Disclosure**

We strive to be as transparent as possible regarding our methodology. However, due to proprietary considerations and company confidentiality policies, we are unable to disclose further specifics about the training dataset composition or more granular details of the training setup at this time. We appreciate the understanding of the research community and hope that the provided information is sufficient to contextualize our findings and facilitate the reproducibility of our core concepts. We are committed to contributing to the open exchange of scientific knowledge within the bounds of these constraints.

