# OpenReview forum: "AdaCoT: Pareto-Optimal Adaptive Chain-of-Thought Triggering via Reinforcement Learning"
_NeurIPS.cc/2025/Conference — Submitted to NeurIPS 2025_

### Official Review · Reviewer_R4k9 · 2025-06-27

**Clarity:** 3
**Significance:** 3
**Originality:** 2
**Rating:** 4
**Confidence:** 4

**Summary:**

This paper introduces AdaCoT, a reinforcement learning-based framework that enables large language models (LLMs) to dynamically decide when to use CoT reasoning based on the queries. By formulating adaptive reasoning as a Pareto optimization problem, AdaCoT balances performance and computational costs, significantly reducing unnecessary CoT steps while maintaining accuracy on complex tasks. And experiments show AdaCoT achieves Pareto frontier.

**Questions:**

[1] The experiments of AdaCoT framework mostly relies on internal models. Could the authors provide the performance of applying AdaCoT to other open-source models, and analyze the impact of different base models on the framework?

[2] What is the intuition of the design of SLM, or why does excluding the loss of the token immediately succeeding the <think> tag help? Are there any theoretical or empirical explanations?

[3] How does AdaCoT handle ambiguous or adversarial queries, like questions that seem simple but requre implicit reasoning? Are there failure cases where it incorrectly skips CoT, and what is the frequency?

**Ethical Concerns:**

["NO or VERY MINOR ethics concerns only"]

**Final Justification:**

The authors' rebuttal has addressed most of my concerns. However, the performances of applying AdaCoT to other open-source models are not provided explicitly with numerical results, still leaving concerns about the generalizability. Thus, based on current results demonstrated in the paper, I lean towards boardline acceptance.

**Limitations:**

yes

**Quality:**

2

**Strengths And Weaknesses:**

**Strengths**

[1] Firstly, this paper is well written. By framing adaptive reasoning as a multi-objective optimization problem, AdaCoT successfully balances the response accuracy and the deployment costs on different tasks, demonstrating its efficiency in LLM reasoning.

[2] This paper presents rigorous experiments across 15 benchmarks and real-world production data, showing significant computational savings of AdaCoT while maintaining competitive performance compared to Full CoT baselines.

**Weaknesses**

[1] The experiments in this work are only conducted on "internal models", which raises concerns about the generalizability and reproducibility of AdaCoT to other open-source LLMs. While the proposed framework is theoretically model-agnostic, its dependency on undisclosed models makes it challenging to assess whether the observed significant gains stem mainly from AdaCoT or model-specific traits.

[2] As acknowledged by the authors, the binary CoT triggering mechanism (on/off) may be suboptimal for queries requiring intermediate levels of reasoning. The authors might could discuss how much the fine-grained control over reasoning depth would influence the Pareto frontier.

[3] The principle-guided CoT labeling relies on an auxiliary internal model's judgement, whose biases or limitations may propagate to AdaCoT's training data. Although it is recognized that automated labeling is more feasible and scalable, the potential biases in training data still remains concerns about its impact on the final model.

---

> ### Author Rebuttal · Authors · 2025-07-31
>
> We are sincerely grateful to you for your insightful comments and valuable questions. This feedback is instrumental in strengthening our work. We address each of the points below.
>
> Response to Weaknesses
>
> Regarding Generalizability and Experiments on Internal Models (also addresses Question 1):
> We sincerely appreciate the reviewer's concerns about the generalizability and reproducibility of AdaCoT. We are actively preparing to conduct related experiments on open-source baselines. For this submission, we have strived to make our methodology as easy to follow as possible, providing extensive details on training parameters and data annotation prompts in the appendix.
>
> We would also like to humbly clarify the primary focus of our work. AdaCoT aims to introduce a novel framework that can control the CoT triggering decision boundary by simply adjusting an RL penalty coefficient, without needing to alter the training data. While the method is still in its early stages, our main goal was to explore how to achieve a better balance between a model's reasoning efficiency and its performance, rather than to directly compare performance across different models. Although we were unable to include experiments on open-source models at this time, we hope our findings offer valuable insights to the community.
>
> Furthermore, to test the generalizability of our strategy internally, we have successfully adapted the same policy to vision-language models of varying scales (a 2.5B/25B MoE pair and a 23B/230B MoE pair). We found that our strategy adapts well to models of different sizes and, surprisingly, is also effective for visual understanding and question-answering tasks.
>
> Regarding the Binary (On/Off) CoT Mechanism:
> We agree with the reviewer that fine-grained control over reasoning depth is a valuable direction for future work. Indeed, our current work represents an initial exploration focused on a binary (trigger vs. not-trigger) control mechanism. However, our framework is inherently extensible to a multi-level CoT scenario, such as distinguishing between "no CoT," "short CoT," and "long CoT." This could be achieved by adapting the data annotation to include these multiple levels and employing a corresponding penalty matrix during the RL phase to manage the transitions between different reasoning depths.
>
> Regarding Potential Bias in the CoT Labeling Process:
> This is a crucial point regarding potential label bias from the auxiliary model. To validate our automated labeling process, we had three human experts review and vote on a sample of user prompts. The results showed that the labels generated by our model achieved an accuracy of over 86% compared to the human majority vote. We also observed that the inter-annotator disagreement rate among the human experts was low (less than 30%), likely because the difficulty of most daily user queries is relatively distinct.
>
> Moreover, our method itself is designed to be robust to the initial difficulty labels. As demonstrated in Figure 1 of our paper, we can adjust the model's final CoT triggering boundary simply by tuning the penalty coefficient during the RL stage, without re-labeling the SFT/RL data. Therefore, unless the initial data labeling introduces a significant domain bias, potential inaccuracies are largely addressable during the RL optimization phase.
>
>
> Response to Questions
>
> Regarding Performance on Open-Source Models:
> This question is addressed in our response to Weakness 1 above.
>
> Regarding the Intuition and Design of the SLM:
> Thank you for this question, which allows us to clarify the motivation behind the Selective Loss Mask (SLM). The intuition for this design choice stems from the specific conditions of the RL-Math stage. In this phase, the prompts are uniformly difficult (AIME-level math problems). This can cause the model to learn a high probability for the entire correct response sequence, which in turn drastically suppresses the probability of the non-CoT path (i.e., generating `</think>` immediately after `<think>`). This can lead to a "collapse" of the decision boundary, where the model almost never explores the non-CoT option.
>
> By excluding the loss for the single token immediately succeeding the `<think>` tag, we effectively prevent this decision boundary from collapsing. This intervention has a negligible impact on the training of the reasoning process itself but is critical for ensuring that the model continues to explore both triggering and non-triggering CoT paths during training.
>
> Regarding Ambiguous or Adversarial Queries and Failure Cases:
> This is an excellent question about the model's handling of nuanced queries that may appear simple but require implicit reasoning. We believe the meta-reasoning approach discussed in Section 4.2 could be helpful in addressing such cases. Additionally, it is worth noting that the auxiliary model we use for data labeling is itself a powerful reasoning model. Its judgment of a prompt's difficulty is not based on a simple surface-level analysis but is the result of its own reasoning process, which may include attempting to solve the problem.
>
> Nevertheless, we acknowledge that our current model can still incorrectly skip CoT for such ambiguous queries. Systematically identifying and handling these failure cases is a challenging and important open problem. We believe this is a very promising direction for future research.

---

> ### Author Response · Authors · 2025-08-05
> **Following up on our rebuttal**
>
> Dear Reviewer,
>
> We are writing to follow up on our recent rebuttal. As the discussion period is concluding soon, we wanted to make sure we had the opportunity to address any concerns.
>
> Please let us know if our rebuttal successfully addressed your points, or if there is anything at all we can clarify for you.
>
> We look forward to hearing from you.
>
> Thank you for your time and consideration,
>
> The Authors

---

### Official Review · Reviewer_3tLA · 2025-07-02

**Clarity:** 3
**Significance:** 2
**Originality:** 2
**Rating:** 4
**Confidence:** 4

**Summary:**

This paper presents an AdaCoT framework to enable LLMs to adaptively decide when to invoke a long chain-of-thought through reinforcement learning. The paper instructs LLMs to label training queries as either likely benefiting from CoT or likely suitable for a direct answer automatically. Based on the labels, AdaCoT uses SFT and RL to LLMs to assess the complexity of each query internally and decide when to use CoT reasoning. A masking strategy is proposed to avoid the decision boundary collapse when training data is imbalanced. Experiments are conducted on internal 15B/150B MoE LLMs to verify the method's effectiveness.

**Questions:**

-	The CoT usage is measured by the CoT triggering rate. Could other metrics, e.g., the tokens used in the response, be an alternative measurement? Because the LLMs can also generate long chain-of-thought outside the <think></think> tokens.
-	Could you please conduct an additional experiment that uses ground truth ‘needed CoT’ labels derived from LLMs to instruct LLMs to respond to open-source benchmarks? The reported performance can be used to verify the correctness of the automatic labeling process.
-	Could you please provide the comparison with a naïve method that uses a simple classification head to determine whether the CoT is needed or not?
-	In the web applications of ChatGPT and DeepSeek, there is a button that lets users decide whether to enable long chain-of-thought or not. What’s the advantage of allowing LLMs to decide by themselves? Is it possible for the LLMs to make better decisions than humans?

**Ethical Concerns:**

["NO or VERY MINOR ethics concerns only"]

**Final Justification:**

The authors have resolved my concerns, I will raise my rating to 4.

**Limitations:**

No. The methods should be verified on open-sourced LLMs to confirm the universality. The quality of the automatic labeling process should be further tested.

**Quality:**

2

**Strengths And Weaknesses:**

Strengths:
-	Teaching LLMs to differentiate complex and simple questions is a valid research topic.
-	The writing is clear, and the proposed method is intuitive.
-	Experiment results show the effectiveness of AdaCoT, which significantly cuts down CoT rate and token usage without a significant decrease in task performance.
Weaknesses:
-	The proposed method is only verified on internal models, which affects the reproducibility of the work. The trained LLM is not directly compared with other open-source LLMs regarding task performance.
-	In line 119, queries are labeled as either likely benefiting from CoT or likely suitable for a direct answer. The correctness of the labeling process should be verified, e.g., the overlap ratio of the queries labeled as needing CoT by human annotators and LLMs. Besides, being labeled as needing CoT by both humans and LLMs doesn’t necessarily mean the query indeed needs CoT.
-	AdaCoT uses the </think> token to signal whether to use CoT. The LLMs learn to predict </think> after <think> directly if no CoT is needed. Thus, a direct baseline that should be compared is to use a simple classification head on the representation of the <think> token from different LLM layers.

---

> ### Author Rebuttal · Authors · 2025-07-31
>
> We sincerely thank you for your thorough and insightful feedback, which has been invaluable in helping us refine our work. We address the weaknesses and questions raised below.
>
> Response to Weaknesses
>
> Regarding Reproducibility and Comparison with Open-Source Models:
> We sincerely appreciate the reviewer's concern regarding the reproducibility of our work on open-source models. We are actively preparing to conduct related experiments on public benchmarks. For the current paper, we have made every effort to make our method easy to follow, and we have provided as many details as possible regarding training parameters and the prompts used for data annotation in the appendix.
>
> We would also like to humbly clarify the core focus of our work. The primary contribution of AdaCoT is to introduce a method that can control the CoT triggering decision boundary by simply adjusting a penalty coefficient during the RL phase, without altering the training dataset itself. While the method is still in its early stages and has limitations, our goal was not necessarily to outperform other models in absolute performance, but rather to explore a new way to achieve a better balance between a model's reasoning efficiency and its performance. Although we were unable to experiment on open-source baselines for this submission, we hope our method can still share valuable insights with the community.
>
> Regarding the Correctness of the CoT Labeling Process:
> This is an excellent point. To verify the quality of our automatic labels, we asked three human expert annotators to review a sample of user prompts and vote on whether each prompt should trigger CoT. The results showed that the labels generated by our model achieved an accuracy of over 86% when compared to the human majority vote. We believe this high agreement is because the difficulty of most daily user queries is relatively distinct. The inter-annotator disagreement rate among the three human experts was also low (less than 30%).
>
> Furthermore, our method itself is highly robust to the initial difficulty labels. As shown in Figure 1 of our paper, we do not need to re-label the SFT/RL data to adjust the model's final behavior. Instead, we can simply modify the penalty coefficient during the RL phase to shift the CoT triggering boundary. Therefore, unless the initial data labeling introduces a strong domain bias, any inaccuracies can be effectively managed during the RL stage.
>
> Regarding the Comparison with a Simple Classification Head:
> We thank the reviewer for this suggestion. We respectfully argue that a direct comparison with a simple classification head may not fully capture the contribution of our method. The goal of AdaCoT is not merely to achieve high accuracy in a binary classification task ("needs CoT" vs. "not"). Instead, it is designed to provide a flexible mechanism to navigate the Pareto frontier of performance and efficiency. By adjusting a single penalty coefficient during the RL phase—without retraining or re-labeling—we can generate a spectrum of models with different reasoning behaviors. This offers a level of dynamic control over the trade-off that a fixed classifier baseline would not provide.
>
>
> Response to Questions
>
> Regarding Alternative Metrics for CoT Usage:
> This is a very insightful question. The reviewer is correct that a model could potentially "hack" the reward model by generating longer, CoT-like content in its final answer to gain a higher score. However, we did not observe this phenomenon in our experiments. We believe this is for two main reasons:
> Our most difficult tasks are concentrated in the RL-Math stage (AIME-level problems). In this phase, we only used a Selective Loss Mask (SLM) to maintain the decision boundary, without other complex reward penalties. This allowed the model to freely explore and enhance its reasoning capabilities without being incentivized to hack the reward.
> In the RL-General stage, which uses more common daily prompts, we specifically adjusted our reward model to penalize CoT-like patterns appearing in the final answer section.
>
> To further illustrate the impact on response length, we provide a comparison of the token counts for the answer part between our baseline (always-on CoT) and RL-exp2. As shown below, RL-exp2 uses slightly fewer tokens in the answer, suggesting it does not generate unnecessarily long responses.
>
>
> Full CoT RL Baseline
> Dataset, Total Token Num, CoT Token Num, Answer Token Num \
> AIME 2025：10251, 9544, 707 \
> MMLU pro：1973, 1347, 626 \
> simple QA：703, 593, 110
>
> RL-exp2
> Dataset, Total Token Num, CoT Token Num, Answer Token Num \
> AIME 2025：10285, 9590, 695 \
> MMLU pro：2015, 1380, 635 \
> simple QA：680, 575, 105
>
> Regarding an Experiment Using "Ground Truth" Labels:
> This is a very valuable suggestion to directly measure the effectiveness of our labeling strategy. Following the reviewer's advice, we conducted this analysis. We used our labeling policy to annotate three test sets and then evaluated the performance of our RL-exp2 model by forcing it to use or not use CoT based on these labels. The results are compared against the full-CoT performance reported in the paper.
>
> Dataset, Labeled as "Needs CoT", Score with Labeled Strategy, Full CoT Score (from paper) \
> AIME 2025：100%, 72.0, 70.0 \
> MMLU pro：41%, 82.7, 85.2 \
> simple QA：2%, 11.3, 12.2
>
> The results are illuminating. For the very difficult (AIME) and very simple (simple QA) datasets, our labeling strategy is quite accurate. The slight score drop on simple QA is likely influenced more by the base model's capabilities. However, for the moderately difficult MMLU pro set, using the static labels leads to a noticeable performance drop, even below the results of our adaptive RL models in the paper. This strongly suggests that during the RL-General phase, the model learns a more nuanced and self-aware CoT triggering policy that is better adapted to its own capabilities than what the simple binary labels can provide. This is a promising direction for future work.
>
> Regarding the Comparison with a Naïve Classification Method:
> (This question is very similar to Weakness 3, so the response is consolidated there to maintain clarity and avoid repetition.)
>
> Regarding the Advantage of LLM Self-Decision over User Control:
> This is an excellent question about the practical user experience. In our live applications, we actually provide users with three distinct modes, implemented as described in Section 4.3: "Always-on CoT," "Always-off CoT," and our "Adaptive CoT." This allows users to choose the mode that best suits their needs.
>
> From our user data analysis and surveys, we found that most users ask relatively simple questions in their daily use but occasionally have complex queries that require deep reasoning. The advantage of an adaptive model is that it provides a seamless "set it and forget it" experience. Users can receive fast, direct answers for simple questions without long waits, while the model automatically engages in deeper reasoning for difficult problems, providing more reliable answers. This removes the cognitive load and friction of having to manually decide which mode to use for every single query, thus improving the overall user experience.

---

> ### Author Response · Authors · 2025-08-05
> **Following up on our rebuttal**
>
> Dear Reviewer,
>
> This is just a gentle follow-up regarding our rebuttal. We hope our response was helpful in addressing your comments.
>
> We would greatly appreciate your feedback on our response and are eager to answer any remaining questions you might have. Your insights are very important for us to improve our work.
>
> Sincerely,
>
> The Authors

---

### Official Review · Reviewer_teS2 · 2025-07-03

**Clarity:** 4
**Significance:** 4
**Originality:** 3
**Rating:** 5
**Confidence:** 3

**Summary:**

This paper introduces AdaCoT, an RL approach that frames the “generate CoT or not” decision as a Pareto trade-off between accuracy and inference cost. A PPO policy—penalized for missed or redundant reasoning and stabilized with selective-loss masking—learns when to emit CoT. On 15 benchmarks, AdaCoT matches always-CoT accuracy while cutting reasoning traces by 30–60%. In production, it lowers CoT calls to 3–13% and trims token usage by ~69%, without sacrificing complex-task performance.

**Questions:**

1. Could multi-level CoT length (not just on/off) be learned under the same Pareto framework?

2. How well does the trigger policy transfer to a smaller or multimodal base model without re-training?

3. What happens when the reward model is noisy or domain-shifted—does SLM still prevent collapse?

**Ethical Concerns:**

["NO or VERY MINOR ethics concerns only"]

**Final Justification:**

My current rating is mainly based on the effectiveness of the proposed method, as demonstrated by experimental results on a production traffic test set.

**Limitations:**

yes

**Quality:**

3

**Strengths And Weaknesses:**

**Strengths**

1. Propose a principled formulation to cast adaptive reasoning as explicit multi-objective (Pareto) optimisation.

2. Experiments show substantial real-world savings: up to 69 % fewer tokens and over 95 % fewer CoT calls.

**Weaknesses**

1. The binary on/off trigger ignores variation in CoT length — a pivotal factor for truly adaptive reasoning.

2. The initial “CoT-needed” labels are generated by another LLM, so mislabeled seeds might bias the policy.

---

> ### Author Rebuttal · Authors · 2025-07-31
>
> We are very grateful to the reviewer for your thoughtful feedback and insightful questions. We appreciate that the reviewer recognized the value of our work and has provided clear suggestions for improvement. We will address each point below.
>
> Regarding Weakness 1 & Question 1: Binary on/off trigger and potential for multi-level CoT.
>
> This is an excellent point, and we thank the reviewer for raising it. We agree that truly adaptive reasoning can involve a spectrum of verbosity, and our current binary (on/off) trigger represents a foundational first step in this direction. We positioned our work as an early but crucial exploration into making CoT invocation adaptive and cost-aware.
>
> We are pleased to confirm that the AdaCoT framework is indeed generalizable to a multi-level CoT scenario (e.g., No-CoT, Short-CoT, Long-CoT). This extension would primarily require two modifications:
>
> 1.  Multi-level Data Annotation: The data labeling process would be adapted to assign one of multiple reasoning levels (e.g., `L_0`, `L_1`, `L_2`, `L_3`, `L_4`) to each query, instead of a binary label.
> 2.  Generalized RL Penalty: The penalty term in the RL reward function would be extended to handle transitions between multiple states. Instead of separate penalties for "miss" and "overuse," we can introduce a penalty matrix `A`, where `A[i][j]` defines the cost of generating a response of reasoning level `j` when the target level was `i`.
>
> The reward function from Equation 4 could be generalized as follows:
>
> `R_{multi(x, r)} = R_{base(x, r)} - A[L_{target(x)}][L_{gen(r)}] - γ * P_{fmt(r)}`
>
> Here, `L_{target(x)}` is the target reasoning level for query `x`, `L_{gen(r)}` is the level of the generated response `r`, and `A` is the tunable penalty matrix. The diagonal elements `A[i][i]` would be zero, while off-diagonal elements would penalize under-thinking (`j < i`) or over-thinking (`j > i`) accordingly. This provides a principled way to control the granularity of reasoning within our Pareto optimization framework. We will add this discussion to the Future Work section to highlight this promising direction.
>
> Regarding Weakness 2: Potential bias from LLM-generated labels.
>
> This is a very important and practical concern. We acknowledge that labels generated by an auxiliary LLM can introduce noise. To mitigate and quantify this, we performed a human verification study.
>
> Label Quality Verification: We sampled a set of user prompts and had three human experts vote on the necessity of CoT for each. The labels generated by our principle-guided auxiliary model achieved an accuracy of over 86% against the majority vote of the human experts. We also observed that the inter-annotator disagreement rate among the experts was low (less than 30%), suggesting that for a majority of real-world queries, the need for CoT is relatively unambiguous.
>
> Framework Robustness: More importantly, our framework demonstrates strong robustness to potential label noise. As shown in Figure 1, the SFT stage serves as a "warm-up," but the final decision boundary is primarily shaped by the penalty coefficients (`α1`, `α2`) during the RL stage. We can steer the model to vastly different triggering behaviors without changing the underlying SFT/RL data labels, simply by adjusting these RL penalties. This indicates that the RL optimization can effectively overcome moderate biases present in the initial labels.
>
> While we believe our current method is robust, we agree that optimizing the initial labeling process is a valuable direction for future work.
>
> Regarding Question 2: Transferability to other models.
>
> This is an excellent question about the generalizability of our approach. While a trained policy's weights may not transfer directly without any retraining, we can confirm that the AdaCoT methodology is highly effective across different model architectures and modalities.
>
> We have successfully adapted this same framework to our internal 2.5B/25B MoE and 23B/230B MoE visual language models. In these applications, the framework also demonstrated strong adaptive performance on visual question-answering (VQA) tasks, effectively deciding when to generate detailed visual reasoning steps versus providing a direct answer. This shows that the core principles of AdaCoT are not limited to a specific model size or the text-only domain.
>
> Regarding Question 3: SLM's robustness to a noisy or domain-shifted reward model.
>
> Thank you for this insightful question, which allows us to clarify the precise role of SLM.
>
> The primary purpose of Selective Loss Mask (SLM) is to counteract decision boundary collapse caused by a skewed training data distribution, rather than to address reward model (RM) noise. We introduced SLM specifically for the RL-Math stage, where nearly all prompts are complex and require CoT. Without SLM, the model's policy would quickly converge to always triggering CoT, effectively "forgetting" the adaptive ability learned during SFT. By masking the loss on the decision token (the token immediately following `<think>`), SLM preserves the model's exploration space and prevents the decision boundary from collapsing to a single mode. Its effectiveness is empirically demonstrated in Table 1.
>
> The issue of a noisy or domain-shifted RM is a critical and more general challenge for all RLHF-based methods. In our experiments, we continuously monitored for signs of instability or reward hacking but did not observe significant issues. We attribute this to the robustness of our base model and the quality of our RM. The impact of RM noise is largely orthogonal to the specific function of SLM, and we believe AdaCoT is not uniquely vulnerable to this issue compared to standard RLHF pipelines.
>
> We hope these clarifications and additional details have addressed the your concerns. We are grateful for the opportunity to improve our paper based on this valuable feedback.

---

> > ### Comment · Reviewer_teS2 · 2025-08-02
> >
> > Thanks for your answer.
> >
> > As you mentioned "We have successfully adapted this same framework to our internal 2.5B/25B MoE and 23B/230B MoE visual language models. In these applications, the framework also demonstrated strong adaptive performance on visual question-answering (VQA) tasks .... "
> >
> > I believe that including these results in the revised version will strengthen the paper.

---

> > > ### Author Response · Authors · 2025-08-04
> > >
> > > Thank you for your valuable suggestion.
> > >
> > > We agree that these results will further strengthen our paper. We will be sure to include them in the revised version.

---

### Official Review · Reviewer_poCn · 2025-07-03

**Clarity:** 2
**Significance:** 3
**Originality:** 3
**Rating:** 4
**Confidence:** 3

**Summary:**

This work proposes AdaCoT, a novel method that enables LLMs to adaptively trigger Chain-of-Thought reasoning based on query complexity. AdaCoT optimizes model performance with reduced computational costs using Pareto-optimal reinforcement learning.

**Questions:**

- Lack of clarity on training datasets and experimental details. The paper mentions two RL stages, RL-Math and RL-General, but provides no specific details on the datasets or training procedures used in each stage. Clarifying the exact data sources, domain coverage, and training configurations would help readers better understand how the model is trained and the scope of the evaluation.

- The paper does not include a comparison with relevant prior works, such as Arora et al. [1], which also focuses on reduce omputational costs while maintaining high performance. Including such comparisons would provide a clearer perspective on how AdaCoT stands in relation to existing methods for optimizing reasoning efficiency in language models.

- The paper claims to focus on adaptive reasoning, distinguishing itself from previous works that primarily focus on reducing compute costs. However, there is no empirical evidence provided to demonstrate how adaptive reasoning in AdaCoT leads to better user experience or performance. A deeper analysis comparing the impact of adaptive reasoning on real-world tasks would strengthen the claims.

- Incorporating a human evaluation could provide valuable insights into how AdaCoT improves reasoning efficiency and user interaction, helping to validate the practical benefits of adaptive CoT triggering.




[1] Arora, Zanette, Training Language Models to Reason Efficiently

**Ethical Concerns:**

["NO or VERY MINOR ethics concerns only"]

**Final Justification:**

The authors have adequately addressed my concerns in their response. This paper presents a novel and well-motivated approach that enables LLMs to adaptively trigger chain-of-thought reasoning. The experimental evaluation is comprehensive and convincingly demonstrates the method's effectiveness. I recommend acceptance.

**Limitations:**

yes

**Quality:**

3

**Strengths And Weaknesses:**

Strengths:
- The paper is well-structured and easy to follow.

- The idea of adaptively triggering cot to achieve better performance while controlling computation cost is novel.

- The empirical results in experiments are strong evidence for the effectiveness of AdaCoT.

Weaknesses: see Questions.

---

> ### Author Rebuttal · Authors · 2025-07-31
>
> We sincerely thank you for your time and for providing such constructive and valuable feedback. We are encouraged that the reviewer found our paper well-structured, the core idea novel, and the empirical results to be strong evidence of AdaCoT's effectiveness. We appreciate the insightful questions, which have helped us identify areas for clarification and improvement. We address each of the weaknesses raised below.
>
> Regarding Q1: Lack of clarity on training datasets and experimental details.
>
> We sincerely apologize for the lack of clarity regarding our training datasets. We acknowledge that more detail would enhance the paper's reproducibility. While we are constrained by company confidentiality policies from disclosing the exact composition and sources of our proprietary datasets, we are happy to provide a more detailed overview of the data distribution and purpose for each training stage.
>
> SFT Stage: The data for this initial "warm-up" stage was curated to build foundational capabilities. It comprises a broad mix of open-source and internal data covering diverse domains, including but not limited to: instruction following, code generation, creative writing, general knowledge question-answering, and role-playing. The goal was to equip the model with a wide range of skills before specializing its adaptive reasoning.
>
> RL-Math Stage: This first RL stage deliberately focuses on highly complex mathematical problems, such as those at the AIME level of difficulty. The primary objective here is to robustly elicit and strengthen the model's multi-step reasoning capabilities in a domain where the correctness of reasoning is verifiable.
>
> RL-General Stage: The final RL stage introduces a wide variety of domains that are more representative of downstream user applications. The domain types and data proportions in this stage are broadly similar to those in the SFT stage, ensuring the model can apply its adaptive reasoning skills across general-purpose tasks.
>
> As the reviewer noted, further details on the training hyperparameters, such as learning rates and optimizer settings, are provided in Appendix E. We believe this additional context on the stage-wise data focus clarifies our training pipeline, and we will incorporate this summary into the appendix of the final version to improve clarity.
>
> Regarding Q2: Comparison with relevant prior works (e.g., Arora et al.).
>
> Thank you for pointing out the highly relevant work by Arora et al. [1]. We agree this is an important work in the area of reasoning efficiency, and we appreciate the suggestion to include it. Their approach of using RL to encourage shorter, more concise reasoning paths is indeed a valuable contribution.
>
> In the final version of our paper, we will add a detailed discussion of this work in our Related Work section (Section 5). We will highlight both the conceptual alignment (using RL to improve efficiency) and the key distinction of our approach. We believe this comparison will better situate AdaCoT within the literature and further clarify its unique contribution.
>
> Regarding Q3 & Q4: Empirical evidence for user experience and human evaluation.
>
> We thank the reviewer for these excellent and crucial points. To directly address the need for evidence on user experience and to validate the practical benefits of AdaCoT, we have conducted a new human evaluation study using our balanced model, AdaCoT RL Model Exp2.
>
> For this study, human experts evaluated responses on 100 representative real-world user queries, carefully selected for their practical relevance. In this setting, the adaptive model triggered CoT on approximately 28% of the queries. We performed a pairwise comparison of AdaCoT's responses against both a "Full CoT" (always reason) and a "No CoT" (never reason) baseline. The Good/Similar/Bad (G:S:B) win rates were as follows:
>
> AdaCoT (Adaptive) vs. Full CoT Baseline: G:S:B = 23% : 51% : 26%
> AdaCoT (Adaptive) vs. No CoT Baseline: G:S:B = 45% : 39% : 16%
>
> These results provide strong empirical evidence for AdaCoT's value.
> 1.  AdaCoT overwhelmingly outperforms the "No CoT" baseline (winning 45% of cases), demonstrating that it successfully preserves the critical performance benefits of CoT for complex queries where it is needed.
> 2.  AdaCoT remains highly competitive with the "Full CoT" baseline, with the vast majority of responses being of similar or better quality. The cases where "Full CoT" was preferred were often attributed to the inherent stochasticity of model sampling or very subtle differences identified during the fine-grained side-by-side evaluation, rather than a fundamental failure of the adaptive mechanism.
>
> Furthermore, your point about user experience is vital. Our analysis of user preferences on our platform indicates that for many common, simpler queries, users prioritize response speed and low latency (especially time-to-first-token). A lengthy reasoning process can be perceived negatively in these scenarios. AdaCoT directly addresses this by providing fast, direct answers for the high-frequency, simple questions typical of daily use, thereby improving user satisfaction and reducing the engineering load on our serving infrastructure.
>
> We believe this new human evaluation data provides compelling evidence of AdaCoT's practical utility and its positive impact on user experience. We will add a summary of this study to the paper to strengthen our claims.
>
> Thank you once again for your detailed review. We are confident that by incorporating these changes, we can significantly improve the paper and more clearly articulate the contributions of our work.

---

> ### Author Response · Authors · 2025-08-05
> **Following up on our rebuttal**
>
> Dear Reviewer,
>
> Thank you again for your detailed and constructive review of our paper.
>
> We have submitted our rebuttal and tried our best to address the concerns you raised. We would be very grateful to know if our response has clarified these points or if you have any further questions.
>
> We are, of course, happy to engage in any further discussion.
>
> Best regards,
>
> The Authors

---

> > ### Comment · Reviewer_poCn · 2025-08-05
> >
> > Thanks for the detailed clarification. My concerns have been addressed. And I'm happy to raise my score.

---

### Note · Authors · 2025-08-12

Dear Area Chair and Reviewers,

We are deeply grateful for the time and expertise you have dedicated to reviewing our work. Your insightful feedback has been invaluable in improving the quality and clarity of our manuscript.

We would like to extend our special thanks to the reviewers who participated in the discussion phase. Their thoughtful engagement and constructive suggestions allowed us to refine our arguments and better articulate our contributions. We believe  our submitted materials have clearly addressed their questions, particularly those concerning our experimental validation and the framework's extensibility to multi-level CoT strategies.

In response to the initial concerns from the reviewers, we have incorporated substantial revisions and provided detailed explanations in our rebuttal. For example, we have expanded on the motivation behind our Pareto optimization framework and underscored the necessity of Selective Loss Masking (SLM) for model stability. While we did not have the opportunity for a follow-up discussion to confirm in all cases, we are confident that our submitted materials have thoroughly addressed their initial concerns.

Our work introduces AdaCoT, a principled, reinforcement learning-based framework designed to find an optimal balance on the Pareto frontier of performance and computational cost. We think this adaptive approach represents a significant and practical step toward making powerful LLMs more efficient and accessible for real-world deployment.

Thank you again for your time and careful consideration. We hope you will find our rebuttal convincing.

---

### Decision · Program_Chairs · 2025-09-17

**Decision:**

Reject

**Comment:**

The paper proposes an adaptive triggering mechanism for chain-of-thought reasoning in large language models. The central idea is to formulate CoT as a Pareto optimization problem that balances performance and computational cost. The method employs PPO with penalty coefficients to adjust the CoT triggering boundary and introduces Selective Loss Masking to stabilize training. Experiments across 15 benchmarks and production traffic demonstrate that AdaCoT substantially reduces token usage while maintaining competitive accuracy.

The main strength of the work lies in its practical motivation. The authors show clear empirical improvements in efficiency, including significant token reductions in production settings.

However, the weaknesses are notable. The method is largely heuristic and engineering-driven, with limited scientific depth. The Pareto framing basically reduces to tuning penalty coefficients in an RL reward function. The reliance on LLM-generated labels raises concerns about bias, and the binary on/off triggering overlooks the more nuanced spectrum of reasoning length and style. Moreover, while the experiments are broad, they primarily highlight cost savings rather than providing deeper insights into adaptive reasoning or advances in optimization. The connection to prior work on CoT compression and reasoning efficiency is also not strongly differentiated, making the contribution feel incremental given the rapidly growing literature.

During rebuttal, the authors clarified aspects of training data, offered comparisons to prior work, and included a small-scale human evaluation. However, these additions did not substantially strengthen the scientific case. The method remains primarily heuristic, with novelty residing more in system framing than in theoretical or methodological innovation.

Overall, the work shows that simple adaptive strategies can reduce costs without major performance loss, which is practically valuable. Yet the scientific merit remains unclear, as the contribution is incremental and heuristic rather than conceptually or theoretically deep. Also, please note that equation (3) is not equivalent to the stated multi-objective optimization problem, where necessary conditions on the hyperparameters should be more explicitly provided and discussed.